# The Influence of Diet on Oxidative Stress and Inflammation Induced by Bacterial Biofilms in the Human Oral Cavity

**DOI:** 10.3390/ma14061444

**Published:** 2021-03-16

**Authors:** Ilona Rowińska, Adrianna Szyperska-Ślaska, Piotr Zariczny, Robert Pasławski, Karol Kramkowski, Paweł Kowalczyk

**Affiliations:** 1The Medical and Social Center for Vocational and Continuing Education in Toruń, St. Jana 1/3, 87-100 Toruń, Poland; elfi.irb@autograf.pl (I.R.); dyrektor@spm.edu.pl (A.S.-Ś.); 2Toruń City Hall, Business Support Center in Toruń, ul. Marii Konopnickiej 13, 87-100 Toruń, Poland; p.zariczny@torun.direct; 3Veterinary Insitute, Nicolaus Copernicus University in Toruń, str. Gagarina 7, 87-100 Toruń, Poland; r.paslawski@umk.pl; 4Department of Physical Chemistry, Medical University of Bialystok, Kilińskiego 1str, 15-089 Bialystok, Poland; kkramk@wp.pl; 5Department of Animal Nutrition, The Kielanowski Institute of Animal Physiology and Nutrition, Polish Academy of Sciences, Instytucka 3, 05-110 Jabłonna, Poland

**Keywords:** bacterial complexes, inflammation, periodontitis, oxidative stress

## Abstract

The article is a concise compendium of knowledge on the etiology of pathogenic microorganisms of all complexes causing oral diseases. The influence of particular components of the diet and the role of oxidative stress in periodontal diseases were described. The study investigated the bacteriostatic effect of the diet of adults in in vivo and in vitro tests on the formation of bacterial biofilms living in the subgingival plaque, causing diseases called periodontitis. If left untreated, periodontitis can damage the gums and alveolar bones. Anaerobic bacteria, called periopathogens or periodontopathogens, play a key role in the etiopathogenesis of periodontitis. The most important periopathogens of the oral microbiota are bacteria of all complexes, including the red complex. The obtained results suggest the possibility of using a specific diet in the prevention and treatment of periodontal diseases-already treated as a disease of civilization. The quoted article is an innovative compilation of knowledge on this subject and it can be a valuable source of knowledge for professional hygienists, dentists, peridontologists, dentistry students and anyone who cares about proper oral hygiene. The obtained results suggest the possibility of using this type of diet in the prophylaxis of the oral cavity in order to avoid periodontitis.

## 1. Introduction

### 1.1. Oral Cavity-Its Specificity in Relation to the Human Body

The oral cavity presents a small part of the human body. It is a very special environment, called the “screen” of the whole organism. A specialist in the subject can anticipate microorganism-caused health problems by looking at the oral cavity [1]. Despite its small area, it plays a very important role in the human body complex [2], where it performs several functions. It is the beginning of the digestive system, the respiratory system, it participates in the formation of sounds, it has a protective, taste, secretory and sensory functions, as well as absorption, taking and preparing food for further digestive processes. Oral health significantly determines the quality of human life, determined by the absence of pain, presence of teeth, sufficient amount of saliva, chewing ability and comfort, no symptoms of dysfunction, aesthetic appearance of the face and proper taste sensation [3,4].

### 1.2. Aim of the Work

The goals of our work were:(1)estimation of the influence of food on the induction of inflammation of the soft tissues in the oral cavity in the presence of residual biofilm(2)estimation of the influence of food on the inflammation of soft tissues in the oral cavity in the absence of residual biofilm(3)to estimate the effect of food on supra- and subgingival biofilm: is it the same or different?(4)to analyze the types of food in terms of the possibility of slowing down the development of inflammation in the oral cavity or rejecting the above possibility(5)to estimate which products cause the greatest oxidative stress in the oral cavity of humans.

This article is an attempt to answer all these questions. The oral cavity is the place where a person’s outer world meets his inner world. It has very diverse microbiological conditions that are constantly changing. If we do not take care of proper hygiene of this site, lesions can appear in the oral cavity, which over time contribute to pathological phenomena throughout the body-systemic diseases, and vice versa-all general diseases are reflected in the oral cavity. Physical, chemical and immunohistochemical factors also influence the bacterial environment in the mouth [5]. These are temperature, redox potential (Eh) [6], concentration of hydrogen ions, availability of nutrients, colonization with microflora, pH of saliva [7,8,9,10,11], order of oral cavity colonization or resistance to it [12,13,14,15], conditions of bacterial adhesion to plaque [4,12,16,17,18]. These conditions are constantly changing as food is introduced into the oral cavity, which lowers the pH, favoring the development of carious lesions in the tooth tissues.

Contemporary periodontics and implantology pose many therapeutic challenges due to the multitude of disease processes affecting the structure of periodontal tissues. Bleeding caused by gentle examination with a periodontal probe is a sign of gingivitis. Additionally, the high frequency of replacing missing teeth with dental implants necessitates increased control and periodontal care in both healthy patients and those with reduced periodontium. At the 2018 annual FDI World Dental Federation Congress organized by the American Academy of Periodontology (AAP) and the European Federation of Periodontology (EFP) in San Francisco (CA, USA), a new classification of periodontal diseases and implant-related diseases was presented to help dentists treat them more effectively [19]. The principles of dealing not only with inflammatory forms of periodontal diseases but also with deformations of the gingival mucosa and periodontal diseases are presented. In addition, the most important disease entities from the surgical point of view, which affect the achievement of the best and longest therapeutic effect, were analyzed, regardless of whether in non-surgical or surgical treatment of periodontitis and tissue around the implant [19]. Periodontal diseases, in the light of the new classification, have been divided into three general categories: gum disease, periodontitis, and other conditions affecting the periodontium. The periodontitis category includes:-periodontitis-necrotic periodontal diseases-periodontitis as a symptom of systemic diseases.

Before making a diagnosis, the dentist should consider the patient’s overall health and some of the risk factors, particularly:-smoking-variety of diet-nutritional deficiencies, e.g., vitamin C deficiency-hormonal changes such as maturation-diabetes.

The classification includes new categories based on disease severity, extent, rate of development, and treatment complexity. Thus, periodontitis as a symptom of systemic diseases is divided into four categories taking into account the underlying general disease, according to the International Statistical Classification of Diseases and Related Health Problems (ICD) code:

Stage I: Initial periodontitis 

Stage II: Moderate periodontitis 

Stage III: Severe periodontitis with possible additional tooth loss 

Stage IV: Severe periodontitis with possible loss of dentition

There are three grades based on the patient’s overall health, risk factors, indications or risk of rapid progression, and expected response to treatment:

Grade A: Slow progression 

Grade B: Average rate of progression 

Grade C: Fast rate of progression

Although the stage of periodontitis will remain unchanged, its severity can be increased after periodontal treatment in combination with reliable patient cooperation and effective control of risk factors [19].

For the first time, diseases developing around implants (peri-implantitis) were classified. The importance of examination with a periodontal probe and frequent checkups, which should be the basic conduct of a doctor after diagnosis, was emphasized [19]. However, diseases and conditions around the implant are divided into four categories: -No changes around the implant,-Inflammation of the mucosa around the implant,-Peri-implantitis,-The loss of soft and hard tissues around the implant.

That is why home prophylaxis in the form of prevention and maintenance treatment is so important [19]. The diagnosis of periodontal disease and its severity is only clinical and is performed on the basis of periodontal charts [19,20,21,22,23,24,25,26,27,28,29,30,31,32,33,34,35,36,37,38,39,40,41,42,43,44,45,46,47,48,49,50,51,52,53,54,55,56,57,58,59,60,61,62,63,64,65,66,67,68,69,70,71,72,73,74,75,76,77,78,79,80,81,82,83,84,85,86,87,88,89,90,91,92,93,94,95,96,97,98,99,100,101,102,103,104,105,106,107,108,109,110,111,112,113,114,115,116,117,118,119,120,121,122,123,124,125,126,127,128,129,130,131,132,133,134,135,136,137,138,139,140,141,142,143,144,145,146,147,148,149,150,151,152,153,154,155,156,157,158,159,160,161,162,163,164,165,166,167,168,169,170,171,172,173,174,175,176,177,178,179,180,181,182,183,184,185,186,187,188,189,190,191,192,193,194,195,196,197].

### 1.3. Oral Microflora Colonization

Natural changes in the oral microbiome follow the eruption of deciduous and permanent teeth. Some 1200 bacterial strains have been observed in the oral cavity, but only some of them have been attributed any responsibility for advanced periodontal disease [1,2,3,4]. It is to clear up that the pathogenic role in periodontology does not correspond to a single bacterium or a limited number of periodontopathogenic bacteria. The main features of the ecosystem are that it has very effective mechanical barriers, it is able to carry and dose nutrients and gas. Moreover, it promotes the exchange of recombinant plasmids. Once the ecosystem has been structured, it is almost impossible to change or destroy it, except through professional oral hygiene [20,21,22,23,24,25,26,27,28,29,30,31,32,33,34,35,36,37,38,39,40,41,42,43,44,45,46,47,48,49,50,51,52,53,54,55,56,57,58,59,60,61,62,63,64,65,66,67,68,69,70,71,72,73,74,75,76,77,78,79,80,81,82,83,84,85,86,87,88,89,90,91,92,93,94,95,96,97,98,99,100,101,102,103,104,105,106,107,108,109,110,111,112,113,114,115,116,117,118,119,120,121,122,123,124,125,126,127,128,129,130,131,132,133,134,135,136,137,138,139,140,141,142,143,144,145,146,147,148,149,150,151,152,153,154,155,156,157,158,159,160,161,162,163,164,165,166,167,168,169,170,171,172,173,174,175,176,177,178,179,180,181,182,183,184,185,186,187,188,189,190,191].

It is a highly organized ecosystem and pathogenicity rides on the biological and behavioral host’s features. In the oral cavity, bacteria concentrate into specific groups, otherwise known as complexes, thanks to which they make better use of nutrients and more effectively defend themselves against the defense mechanisms of the macroorganism [20,21,22]. Periodontal tissues in the oral cavity are destroyed by various factors of bacterial or indirect origin as a result of an inflammatory reaction. The cause of direct toxic action are endotoxins, exotoxins, enzymes and end products of metabolic transformations [23]. The best-known toxin is the leukotoxin produced by *A. actinomycetemcomitans* and responsible for the destruction of neutrophils, monocytes, causing cell lysis (increasing the permeability of their membranes). The endotoxins of *P. gingivalis* and *A. actinomycetemcomitans* cause the release of substances such as interleukin 1 beta and prostaglandin E2 by monocytes, fibroblasts and macrophages, which are actively involved in bone resorption [24]. Many bacterial enzymes destroy intercellular substances (for example, collagen) and the connective tissue of the macroorganism. The end products of metabolic transformations of the microorganism (e.g., butyric acid, propionic acid), ammonia, indole, amines, volatile sulfur compounds produced in large amounts destroy the mucosa permeability and are reducers of collagen synthesis [23].

The order of microbial colonization in the oral cavity depends on the availability of individual nutrients and the crossing of the natural limit of non-specific immunity. The first class of microorganisms is carried by the macroorganism and is obtained directly from the environment. It colonizes specific ecological niches and reproduces in real time, creating ecological communities. The environment in which it lives changes under the influence of its metabolic activity, which facilitates the penetration of other species of bacteria into them, starting ecological succession, which leads to the creation of a large and very diverse environment of settled pathogenic microflora in the oral cavity, which begins to induce specific pathological changes. Each introduction of a new non-bacterial element into the oral cavity environment, from the first erupting tooth to the complete prosthesis, contributes to the creation of new environmental conditions dependent on the pH of the saliva, physiology of the mucosa and epithelium covering the entire inside of the cheek. which lowers the oxyreduction potential of a given niche, which creates favorable conditions for the development of anaerobic bacteria. Thus a new autogenous succession is formed. For example the cheek epithelium is mainly inhabited by streptococci—the most common species being *S. sangius*, *S. mitis*, *S. salivarius*, *S. vestibularis*, *S. anginosus*. The bacteria are not regularly present on other surfaces of the oral epithelium. It was determined that there are 5–25 microorganisms per one cell of the cheek epithelium [6,25,26]. 

For the life and development of microorganisms living in the oral cavity the amino acids, proteins, carbohydrates and glycoproteins in saliva are sufficient. Fluids flowing out of the gingival crevices are a rich source of nutrients for bacteria. Another endogenous generator of nutrients are substances triggered by decaying diseased periodontal tissues due to hydrolytic enzymes, i.e., protease, collagenase, hyaluronidase, DNA-ase produced by bacteria in the oral microflora. A significant source of nutrients for the bacteria living in the mouth are the ingredients from everyday food, especially carbohydrates. Carbohydrates are converted into glucans and fructans, and they also contribute significantly to the formation of plaque. The bacteria that live in it, i.e., *S. mutans*, *S. gordonii*, *Lactobacillus*, are responsible for the acidification of the environment, which promotes the demineralization of enamel and dentin [6,7,8,9,10,11,12,13,14,15,16,17,18,19,20,21,22,23,24,25,26,27,28,29,30,31,32]. Other substances are, for example, polypeptides that bind saliva glycoproteins to cells from other bacteria or to calcium ions. Extracellular polypeptides, which react with gluconic proteins of other streptococci, behave in a similar way. Streptococcal cell walls contain lipoteichoic acids that can bind to the acquired salivary sheath. An important role in the colonization processes is played by polyproteins of the cytoplasmic membrane, which act as a “transporter” of components, i.e., sugars or peptides related to the acquired salivary sheath or surfaces of other bacterial cells. So-called ligands-host factors are negatively charged salivary glycoproteins and can be found on mucosal epithelial cells, the hard tissues of the teeth above and below the gums, and on prosthetic restorations. These compounds are positively charged through calcium compounds (Ca^2+^), which bind to negatively charged bacterial adhesins in streptococci like *S. mutans, S. oralis, Actinomyces*, Gram-negative bacteria, i.e., *P. intermedia, F. nucelatum, P. gingivitis, E. corrdens* [25,26,27,28,29,30,31,32,33,34].

### 1.4. The Colonization of Microorganisms on the Tooth Surface

The colonization of microorganisms on the tooth surface begins with specific surfaces and the initiation of the initial stages of microbial adhesion to the surface. This phenomenon is conditioned by many factors, such as adhesins-specific proteins that bind to carbohydrates through structures called fimbria (these are fibers of various lengths and appear on the cell). Thanks to them, lipophilic surfaces of bacterial cells affect and connect with the hydrophobic surfaces of epithelial cells. This contact resembles the binding of an enzyme to a substrate, and receptors on the surface of epithelial cells are recognized by adhesins found on some microorganisms. Fimbriae were first detected in streptococci such as *S. anginosus, S. salivarius, S. oralis, S. mutans*, and then on actinomycetes like *A. viscosus, A. naeslundi* and gram-negative bacilli (*Prevotella, P. intermedia, P. oralis, P. bucceae, P. melaniogenica,* and *Porphyromonas gingivalis* species) [25,26,34]. The first pioneering microorganisms appear in the mouth of a newborn baby after 12–18 h. These are aerobic and relatively anaerobic bacteria, the most common ones being: *S. oralis, S. salivarius, S. mitis*, with time *S. gordonii* and *S. anginosus* join them. Within 1–7 months, the pioneering flora is increasingly diverse, and especially anaerobic Gram-negative anaerobes appear, such as *Fusobacterium*, *Prevotella*, *Veillonella*, sometimes *Capnocytophaga*, *Leptotrichia*, *Campylobacter*, *Eikenella*. There may be 0 to seven species in the mouth. The bacterial flora is enriched with the eruption of primary teeth (1–3 years). There are also species of the genus *Acitinomyces*, *Neisseria, Lactobacillus, Porphyromonas, Rothia and Actinobacillus*. The oral microflora reaches relative homeostasis as the body reaches adulthood, but when specific immune disorders, tumors, or chemotherapy occur, opportunistic microorganisms such as *Klebesiella, Candida, Escherichia, Staphylococcus, Pseudomonas* may appear [15,16,17,18,19,20,21,22,23,24,25,26,27,28,29,30,31,32,33,34,35,36,37,38,39,40,41,42,43,44,45,46,47,48,49,50,51,52,53,54,55,56,57,58,59,60,61,62] (Figure 1).

Another important mechanism of adhesion is bacterial aggregation, based on the attachment of microorganisms to each other, as well as their attachment to various surfaces. The main substrates in the oral cavity are a casing acquired on the hard surfaces of the enamel and root cement (osseous), on the oral mucosa. The sheath covering the surfaces is approximately 1 micron thick and is specific to each surface. The sheaths on the hard tissue surfaces are not identical, on the epithelium they are referred to as the mucous sheath [59,60]. The casing is removed from the abovementioned tissues and regenerated after 90–120 min. It consists of water, lipids, proteins, glycoproteins, mineral salts, and also contains acidic, proline-rich proteins and stearin, which facilitate the adhesion of *S. mutans*, *A. viskosus* and anaerobic bacteria with black pigments. The above phenomenon is related to the occurrence of lectins-carbohydrate-protein compounds showing affinity for sugar residues in cells of other microorganisms. Gram-positive bacteria gather between *S. sanguis* streptococci and *C. matruchotii* or *P. acnes* Gram-positive bacilli. The same applies to Gram-negative rods, i.e., *P. gingivalis, F. nucelatum*, as well as Gram-positive bacteria *Actinomyces*, *Staphylococcus* and Gram-negative bacteria of the genus *Prevotella, Fusobacterium, Capnocytophaga, Eiconella, Porphyromonas* and *Veillonella* [6,25,26,37,38,39,40]. Another important phenomenon in the colonization process are extracellular polymers produced by *S. mutans*. They are produced only in the presence of sucrose and aggregate very quickly on smooth enamel surfaces. The same is true for *Actiniomyces*, but they colonize the subgingival surfaces of the tooth. These processes initiate the formation of dental plaque, which results in carious lesions [41,42,43,44,45,46,47,48,49,50,51,52]. Sometimes the inhibition of colonization processes may stop. The inhibition of colonization processes occurs due to the possibility of stabilizing the bacterial flora. Stabilization occurs largely due to the possibility of eliminating niches of opportunistic and exogenous microorganisms (transitional flora). Often the microorganisms of the transitional flora are pathogenic to the macro organism, and the inability to colonize them is due to the lack of availability of adhesins-nutrients in a populated niche. The important thing for sedentary microflora is that it can produce some antagonists-substances that interfere with the colonization of the niche by strains of exogenous bacteria. It is suspected that this may be due to the hydrogen peroxide produced by *S. mitis* or *S. salivarius* streptococci, which produce salivaricin. They play an important role by inhibiting the development of pyogenic streptococci such as *S. pyogenes* [6,15,25,26,36]. Bacteria such as *S. mutans*, *C. martuchotii*, *A. actinomycetemcomitans* additionally produce enocin (bacteriocin), which is proteinaceous in nature and has the ability to inhibit growth and even kill related organisms. A very important role for the stability of the flora of the sedentary ecological niche is played by the non-specific immunity and acquired immune resistance of the macroorganism, as well as some external factors. Long-term treatment with antibiotics and cytostatics with a broad spectrum of action, which often eliminate oral bacteria and *Candida* yeasts appear in their place. Due to the presence of fibronectrin in epithelial cells, it is not possible to colonize the oral cavity with *P. aeruginosa* or other gram-negative bacteria species [15,36].

### 1.5. Bacterial Biofilm-Oral Health and Inflammation

Bacterial biofilm or plaque is the cause of most oral health problems. The accumulation of bacterial biofilm is a basic factor in the pathogenesis of periodontal diseases, which, together with general factors, affects the defense of the macroorganism and has a significant impact on the extent and dynamics of inflammatory processes. Bacterial plaque is a soft mass that fits tightly to the teeth, it is a cluster of bacteria and an intercellular matrix that forms in the mouth on hard and soft structures. The bacterial plaque adheres tightly to the surface, it cannot be rinsed, even with preparations under high pressure. It can only be removed with a toothbrush, with appropriate movements depending on the conditions in the oral cavity, or with the use of specially designed dental tools [4,43,44,45,46,47,48,49,50,51,52,53,54,55,56,57,58,59]. Clinically, it is a yellowish, consistent mass, slightly darker than the color of the enamel, and covers the teeth in inverse proportion to the host’s oral hygiene. Dental plaque develops in several stages, has no set values, and changes its volume, structure and content if it is not removed. Its composition depends on the location, thickness, physicochemical properties of saliva, as well as the diet. The first stage of its formation is the acquired casing called pellicula, which is formed immediately after cleaning the surface of the teeth and is about 1 micron thick. The surfaces are covered with saliva glycoproteins, which form a protective layer on the tooth and at the same time create favorable adhesive conditions for bacteria. A few hours after brushing your teeth, early plaque forms. After about 3 h, the bacteria colonize the plaque to form large colonies of aerobic bacteria, such as *Streptococcus sanguis* (they like to settle in the organism on valves), *Streptococcus mutans*, *Streptococcus mitis* (responsible for infective endocarditis), etc. Anaerobic bacteria with a unique ability to invade tissue by the production of enzymes and toxins (responsible for gingivitis) and *Porphyromonas gingivalis* contributing to platelet aggregation, which in turn leads to the formation of blood clots in blood vessels. Ripe plaque—in clinical form—is detectable after 24 h. As the plaque matures, the number of anaerobic colony-forming bacteria increases sharply. Not removed for 7–10 days, it is visible on the teeth with the naked eye. It contains 70% of microorganisms (anaerobic bacteria, fungi, viruses) and 30% of an organic matrix-the “skeleton” that binds these components together [6,7,8,9,10,11,12,13,14,15,16,17,18,19,20,21,22,23,24,25,26,27,28,29,30,31,32,33,34,35,36,37,38,39,40,41,42,43,44,45,46,47,48,49,50,51,52,53,54,55,56,57,58,59,60,61,62].

Plaque microorganisms persist on a matrix composed of polysaccharides (elements of bacterial origin) and organic substances of the host-proteins, carbohydrates, food debris, dead cells from oral tissue, red and white blood cells, enzymes and toxins (bacterial products), glycoproteins and saliva antigens, which take part in the so-called early colonization process, occur in the plaque up to day 7, and after 7 days the late colonization and maturation of plaques begins [6,18,19,25,26,37,38,39,40,60,61,62].

If it is not removed, the plaque mass begins to mature, sticking to the clean surface of the tooth. If the oral cavity is not properly cleaned or not cleaned at all due to its specific environment, it provides the microorganisms living in it with excellent conditions for growth. There are approximately 2 × 10^11^ microorganisms in one gram of plaque. In addition, there are many factors in the oral cavity that contribute to the retention of plaque while preventing its effective removal. It is favored by large or uneven fillings, bridges, prosthetic crowns, implants, removable dentures, orthodontic appliances, gingival pockets, tooth crowding, malocclusion, fixed or removable dental splints. Bacterial plaque covers the hard structures above and below the gums in the mouth, so it is divided into plaque and subgingival plaque. This division is justified because these structures are inhabited by various microorganisms and differ in their surroundings, which is favored by the appropriate pH [10,46,47,58,63,64]. A supragingival plate is placed over the tooth crowns at the gingival margin and forms a soft, yellow-white mass. It mainly inhabits the edges of the gums, the area of suspended fillings, prosthetic works, furrows, etc.-places that are difficult to clean. The speed of its formation is very individual and depends on parameters such as diet, salivation and oral hygiene. Small amounts are usually invisible to the naked eye but can be detected with a probe or a preparation that stains the bacterial biofilm. In the supragingival plaque, the matrix constitutes 50% of its mass, and the bacterial flora is mixed with the majority of aerobic bacteria. The metabolism of the supragingival plaque results from carbohydrate metabolism [10,46,47,58,63,64]. The subgingival plate is located in the gingival groove or in the periodontal pocket (less than 3 mm of the gingival margin). If not removed from the tooth surface, it develops towards the gingival groove. Its presence can be detected with an instrument or, if possible, by removing the gingival margin. The composition of the supra- and subgingival plaque is not the same due to the condition in the gingival sulcus. Anaerobic bacteria grow rapidly and colonize such as *Porphyromonas gingivalis*, Gram-negative rods and spirochetes. In the gingival groove, the effectiveness of the cleansing mechanisms is limited, there is a different redox potential, and the gingival groove facilitates the inflow of nutrients for the bacteria. It contains mainly (over 90%) of anaerobic bacteria (there are many of them), and also mobile bacteria [18,19,37,38,39,40,60,61,62]. Unfortunately, the bacterial plaque is not removed under the calcification process and then it can no longer be removed with a toothbrush, but only with manual and mechanical tools in the dentist’s office. The mineralization process takes place with the participation of calcium and phosphorus ions present in the saliva. The calcium phosphate crystals grow in the plaque matrix and enlarge until the plaque is mineralized. Over time, the stone ages as a result of unlimited crystals mixing-first the brusite is formed, then the eight-calcium phosphate, and the mature stone is composed of hydroxyapatite and tricalcium phosphate crystals. The stone crystals grow in close contact with the tooth surface and the stone obtains mechanical retention due to the irregularity of the surface. On the outside, tartar is covered with non-mineralized bacterial plaque and, like dental plaque, it is divided into supra- and subgingival [45,65,66,67,68,69,70,71,72,73]. The supragingival stone is yellow-white in color. It can be the color of the teeth and its smaller deposits are visible after drying the surface in the light of the reflector, the stone differs from the tooth surface by the lack of gloss-a matte structure. On the gingival margin and on the crown, it is often discolored as a result of smoking, consumed food and drinks with strong dyes. The formation of calculus is influenced by the concentration of calcium and phosphorus ions in saliva and the secretion of saliva from the salivary glands, hence it is most often formed at their outlet [7,8,9,10,11]. The subgingival calculus is formed apex to the gingival margin. Most often it is dark because it becomes stained with erythrocytes. It is visible after drying the gingival margin, it shines through the soft tissues, you can gently lift the gingival margin and see it, and also during periodontal surgery. To detect it in deeper layers, you can use a periodontal probe, take an X-ray, use laser equipment [69,70,71], (Figure 2).

It is the bacterial flora that is largely responsible for the formation of plaque, periodontal disease, the occurrence of halitosis (unpleasant odor emanating from the mouth), and in the later stages of macro-organism health problems [48,74,75]. 

### 1.6. Inflammation in the Oral Cavity

Inflammation is one of the body’s main defenses to noxious agents-physical, chemical, or biological (immune and non-immune). They are a complex process of the body’s response to external or internal damaging (pathogenic) factors. The inflammatory process by damaging tissues and organs can get out of hand, leading to undesirable outcomes. Inflammation is a process that occurs at certain stages. The role of the body’s defenses is primarily to remove the inflammatory factor, repair damaged tissues, and protect against the development of the disease-that is, the protective function. Chronic inflammation can evolve into acute inflammation, becoming a pathogen that leads to autoimmune diseases and cancer. The inflammatory reaction activates the immune system to fight pathogens (e.g., viruses, bacteria), without it we would not be able to survive even the smallest infection. In the event of an inflammatory reaction, changes in the blood vessels always appear. The blood vessels widen and their permeability increases, leading to mediators and inflammatory cells entering the surrounding tissues [76,77,78,79,80,81,82]. 

Symptoms of inflammation include pain, redness, swelling, warmth, loss of function, and tissue damage, meaning the organ is no longer functioning as it should. The inflammatory response is closely related to the body’s immune response. It begins with the contact of the pathogen with specialized cells of the immune system-antigens. The stimulated antigens produce and release inflammatory mediators that initiate and maintain the inflammatory process. Mediators exert pro-inflammatory and anti-inflammatory effects on target cells by modulating the course of inflammation. Over time, the adaptive immune system (specific response) is also involved in destroying tissue damaging factors. This system works extremely precisely-it cooperates with T and B lymphocytes, leading to the production of specific antibodies that selectively neutralize the pathogen [76,77,78,79,80,81,82]. The inflammatory reaction in the body has three phases-pathogen recognition, pathogen elimination and extinction of the immune response. It should be mentioned that a properly functioning immune system effectively recognizes pathogens and effectively removes them without damaging its own cells and tissues, but in some situations the immune mechanisms may malfunction and an inflammatory reaction to its own antigens occurs. This situation occurs in many autoimmune diseases, such as type 1 diabetes, rheumatoid arthritis, celiac disease, lupus erythematosus, Hashimoto’s disease, etc., [19,83].

In the body’s response to inflammation, in the first phase, in response to the intrusion of microbes into the body’s tissues, many soluble chemicals are produced that kill invading microbes. These substances are lysosomes, interferons, interleukins, other cytokines, acute phase proteins and the complement system. In the next stage, there are cells involved in the phagocytosis, digestion and destruction of foreign particles, i.e., macrophages, neutrophils, eosinophils, monocytes. Additionally, basophils, mast cells, platelets and natural killer (NK) cells are involved in the non-specific immune response. The nature of the infiltration of inflammation depends on the type of microbes that cause it. The inflammatory reaction facilitates the induction of an immune response as a result of the influx of lymphocytes into the inflammatory focus. Certain products of microbial metabolism (for example, endotoxins) may have an adjuvant effect [21,84].

Human mouth infections may affect the gums, alveolar bones, periodontitis, root inflammation (periodontitis) and soft tissues of the tooth. Inflammation appear as a consequence of subsequent phenomena caused by the imbalance of the oral cavity bacterial flora. They are initiated by the remaining plaque. The pathogenesis of gingivitis and periodontitis is similar, but the two entities are described separately. In the case of gingivitis, the bacterial plaque accumulates on the gingival margin, and its bacterial products-metabolic by products, i.e., proteases, H_2_S, endotoxins penetrate the epithelium and initiate gingivitis. The answer to the above situation is increased permeability and dilatation of blood vessels, and leakage of fluid into the tissues and the gingival groove [76,77,78,79,80,81,82]. Neurophiles from the gingival groove and blood vessels begin to migrate. As a result, the collagen fibers surrounding the blood vessels break down towards the apex. After a few days, lymphocytes (especially T-type) and macrophages begin to accumulate, and fibroblasts, due to morphological changes, have a lower ability to produce collagen. As a result, plasma cells (so-called inflammatory cells) begin to dominate. Collagen and the expanded connective epithelium are still destroyed. Inflammatory (chronic) gingivitis is confined to adjacent tissues with the connecting epithelium and the gingival groove epithelium. Gingivitis is defined as baseline, early and established. Initial inflammatory changes develop within 0–4 days with plaque build-up and are characterized by vascular permeability, migration of neurophiles out of the vessels into the tissues, and damage to the collagen fibers surrounding the blood vessels. Early inflammatory lesions will develop 4 to 7 days after plaque build-up. It leads to further processes that started in the initial changes, moreover, during this period, lymphocytes (especially T-type) and macrophages begin to accumulate, the cytotoxicity of fibroblasts changes, which is manifested by a decrease in collagen production and proliferation of epithelial cells. The so-called fixed inflammatory lesion becomes apparent from about 14–21 days without disturbing the plaque growth. This coincides with the clinical diagnosis of chronic gingivitis. In the previous steps, further changes that cause the erythematous gingiva to dark blue and become the dominant inflammatory cells were discussed. Inflammatory lesions are confined to the gums, not to the alveolar bone. They can be modified by general factors: those related to the endocrine system (gingivitis, inflammation related to the menstrual cycle, pregnancy gingivitis, gum disease in the course of diabetes), blood diseases (gum leukemia), drug-modified (anticonvulsants, immunosuppressants, channel blockers) and even contraceptives) and modified by an eating disorder [3,6,36].

It should be noted that there are gum diseases that are not associated with plaque. These include diseases related to infections (bacterial, viral, fungal), genetic conditions (gingival fibromatosis), the course of systemic diseases (desquamative gingivitis, e.g., with lichen planus, pemphigus, bullous dermatosis, erythema multiforme) and traumatic origin (as a result of own actions, by a doctor, erosions, ulcers or recession). In a situation where the above changes spread and begin to affect bone tissues, they will coincide with the clinical diagnosis-periodontitis. However, chronic inflammation can last a long time. It is suspected that there is an important balance between the microbes and the host response in this situation. Reducing the host’s immunity or increasing pathogenic flora can tip the scales in favor of periodontitis, and the inflammatory lesions then spread to the bone tissue, i.e., alveolar bone, periodontitis, and root cementum. Inflammatory lesions are characterized by dilatation and proliferation of blood vessels, plasma cells and B lymphocytes in connective tissue, the pocket epithelium becomes thinner, it is often ulcerated and leaks bacterial products, inflammatory mediators of defense cells. The degradation of connective tissue is evidenced by visible necrotic foci. Connective epithelium proliferates in the apical direction [76,77,78,79,80,81,82]. Cement is exposed, absorbing bacterial products, it becomes soft and necrotic. Bone resorbs prostaglandin endotoxins, interlequins and tumor necrotic factors (TNF). Periodontitis manifests itself as chronic inflammation of the alveolar bone of the tooth suspension apparatus [85,86,87]. Responsible for specific bacteria or groups of bacteria. As a result of their activity, the fibers of the alveolar and periodontium bones are destroyed, which results in the formation of periodontal pockets and gingival recessions (often occurring simultaneously), tooth mobility, changes in the consistency and shape of the gums and bleeding during probe examination. Frequent bleeding during examination indicates the presence of inflammation and loss of connective tissue attachment, which may be continuous or episodic [76,77,78,79,80,81,82,83,84,85,86,87].

There have been several classifications of periodontal diseases in recent years and the current one is quite simplified. It begins with gum disease (already outlined briefly), through chronic periodontitis, aggressive periodontitis, periodontitis in the course of general diseases, acute periodontitis, to congenital or acquired defects [88].

Chronic periodontitis is the most common disease found in adults, but it also occurs in children and adolescents. The main cause of it is a bacterial infection, most often provoked by *Porphyromonas gingivalis*, *Bacteroideforsythus*, *Tremponema denticola*, *Actinobacillus actinomycetemcomitans,* or less frequently by *Prevotella intermedia*, *Camphylobacter rectus*, *Peptostreptococcus micros*, *Fusobacterium nucleatum* and *Eikenella corrodens* [18,19,32,37,38,39,40,60,61,62].

The criteria for diagnosing aggressive periodontitis include the analysis of clinical, radiological, histological, laboratory tests and, in cases of doubt, microbiological and immunological tests. It is a specific type of periodontitis and concerns the suspension apparatus of the tooth. They are distinguished by characteristic features that can be clearly identified clinically and in a laboratory, and they are characterized by rapid tissue destruction. It occurs in patients with a disturbed immune system. The disease occurs in their families and is much more likely to be caused by *Actinobacillus actinomycetemcomitans* (up to 90%), *Porphyromonas gingivalis*, *Fusobacterium nucleatum*, *Camphylobacter* rectus and others. It can occur in a localized and generalized form [18,19,34,37,38,39,40,60,61,62].

Periodontitis in the course of general diseases, apart from bacterial plaque, is associated with the host response. General factors are responsible for modifying all forms of periodontal disease, as they affect the immune system and also the inflammatory response. The changes in the tissues in the oral cavity are influenced by many diseases and medications, and the diseases affecting the course of periodontitis include: hematological diseases (acquired neutropenia, leukemia and other blood diseases), genetic diseases or syndromes (familial and cyclic neutropenia, Down syndrome), leukocyte adhesion deficiency Syndrome, Papillon-Lefevre syndrome, Chediak-Higashi syndrome, histocytosis, glycogen storage disorders, childhood congenital agranulocytosis, Ehlers-Danlos syndrome, hypophosphatasia) and other systemic diseases (osteoporosis, sex hormones, diabetes and human immunodeficiency virus (HIV) syndrome immunodeficiency, immunosuppression and steroids) [85,89,90,91]. Acute periodontal conditions are characterized by a rapid onset, acute course, severe pain and involve the tissues surrounding the tooth. They can be local or general, with possible systemic symptoms. These include necrotic-ulcerative periodontal diseases (abscesses: gingival, periodontal, acute, chronic periodontal and pericoronal abscesses), endo-periodontal changes, etc., [89,90].

Congenital or acquired defects do not cause the disease but may predispose to gingivitis and periodontitis in the presence of plaque. These include: local odontogenic factors (anatomical structure, tooth positioning, orthodontic appliances, fillings, prosthetic restorations, root fractures, root resorption and hypercementosis), mucogingival deformities within the dental arches (gingival and soft tissue recessions, reduced vestibule depth, abnormal frenulum attachment, gingival overgrowth) and occlusal trauma (primary and secondary) [89,90,91,92,93,94,95,96,97,98,99,100,101] (Figure 1).

### 1.7. Oxidative Stress—A Stimulator of Inflammation in Oral Cavity

Stress is a state of acute tension in the body forced to react and defend against any kind of threat or aggression (infectious, psychological, traumatic, toxic, etc.). Rushing, increased heart rate, muscle contractions are just some of the body’s reactions in a stressful situation. The problem begins when stress becomes chronic, and not reacting to it for too long has a direct impact on health, work, relationships with family and the environment. Psychological stress is associated with the occurrence and progression of periodontal disease. People with high levels of stress and poor coping skills have twice as many periodontal diseases as people with minimal stress and good coping skills. It is known from the literature data that there is a relationship between cortisol values and the degree of periodontal disease. Increased cortisol levels due to stress can affect the immune system and increase the susceptibility of a person suffering from periodontal disease. Excessive stress is often negatively correlated with changes in hygiene and eating habits, increased alcohol consumption, and smoking, which increases the risk of periodontal disease. For these reasons, it can be concluded that, of course, the presence of periodontal pathogens remains an important etiological factor in periodontal disease, although stress control and control will be the key to gingival reduction [102,103,104,105,106].

More studies suggest that maintaining oxidative stress at an appropriate level has a significant impact on the treatment of many inflammatory diseases. It has been studied that the loss of control over oxidative stress blocks the cell cycle, induces cell death and neoplase, causes allergic and autoimmune disorders related to excessive stimulation of the immune system cells and disruption of the immune system’s immune tolerance [22,80,84]. It is already known that oxidative stress is an important etiopathological factor in many systemic diseases and contributes to the faster aging of the body. The list of diseases in which oxidative stress plays a role now includes more than fifty items and is not yet closed. These diseases include: cardiovascular diseases (atherosclerosis, arterial hypertension), neurodegenerative system diseases (multiple sclerosis, Parkinson’s disease), metabolic diseases (obesity, diabetes), neoplastic diseases, eye diseases (glaucoma, cataracts), atopic diseases (rhinitis, asthma), bacterial infections, septic shock, inflammatory diseases (periodontitis, esophagitis, pancreatitis, liver inflammation) [23,82,101].

### 1.8. Oxidative Stress and Periodontitis

Scientific research on oxygen—which we need to live—has shown that it has a disastrous effect on the cells of the body, but its unfavorable effect depend on its availability in the mouth. Thanks to its presence, the following bacteria multiply faster: caprophilic, microaerophilic, obligate anaerobes, and relative anaerobes [15,24,25,33,34,35,59,60,61]. When homeostasis in the oral cavity is disturbed, inflammation appears, the occurrence of which is undoubtedly influenced by oxidative stress. Oxidative stress plays a twofold role, it can be one of the defense mechanisms, but it can also initiate many different pathological changes. Smoking is still not uncommon among people, which leads to the formation of huge amounts of free radicals. There are approximately 1015/35 mL in cigarette smoke (1 puff) and the concentration in cigarette tar is approximately 1017/g. Smoking cigarettes causes an imbalance between proteinases and antiproteinases. It intensifies lipid peroxidation, damages DNA in lymphocytes, lowers the level of some antioxidants in the blood serum, accelerating the process of atherosclerotic plaque formation. As a result, nicotine increases the clinical course of periodontal diseases. On the one hand, the knowledge of the biological properties of reactive oxygen and nitrogen species helps to predict potential dangers in certain disease states. On the other hand, the knowledge of natural antioxidant mechanisms and their use, as well as their support by pharmacological substances neutralizing their negative derivatives, as well as the ability to cope with everyday stress, can be an effective support in the treatment of periodontal diseases in disease states [14,15,16,29,33,36,56,63,78,81,93,97,99,102,103,104,105,106,107,108,109,110,111,112].

The pathognomonic lesion of the periodontium is the periodontal pocket and the modulation of inflammation in the periodontal pocket itself. The modern pathogenetic model of genes, proteins and metabolites in dynamic biological processes is based on a multi-level structure that includes disease initiation and eradication mechanisms, regulated by innate and environmental factors [102,103,104,105,106,107,108,109,110,111,112,113,114,115,116,117,118,119,120,121,122,123,124,125,126,127,128,129,130,131,132,133,134,135,136,137,138,139,140,141,142,143,144,145,146,147,148,149,150,151,152,153,154,155,156,157,158,159,160,161,162,163,164,165,166,167,168,169,170,171,172,173,174,175,176,177,178,179,180,181,182,183,184,185,186,187,188,189,190,191,192]. Polymorphisms in inflammatory genes such as interleukins 17A and 17F, 1, 1B and rs1143634 and MMPs genes are associated with the risk of development of periodontitis [193,194,195,196].

## 2. Diet—The Basis of Human Health and Inflammation

### 2.1. Diet

The world is so organized that every micro- or macroorganism needs food to live. Food is needed for development and growth, it is a source of energy for life, and the nutrients it contains are needed to regulate the processes taking place in the body. In the oral cavity, the role of food does not end there. In addition to the fact that it determines the correct structure of the teeth, the entire stomatognathic system, strengthens the resistance of the tissues in the oral cavity, it can protect against caries, erosion, periodontal disease, but it can also contribute to the formation of caries, erosion and diseases also periodontitis [23,37,48,50,67,75,89,90,91,92,93,94,95,96,97,98,99,100] (Figure 3).

The sources of energy in food are carbohydrates, fats and protein. For the growth and development of tissues, macro- and micronutrients, proteins and vitamins are needed. You need protein, vitamins and, of course, water to regulate your metabolism. Mammalian organisms (including humans) need: water, carbohydrates, fiber, fats, proteins, vitamins, minerals and fiber to function effectively [113,114].

It should be emphasized that vitamins such as vitamin, A, E, C, several B vitamins, flavonoids, coenzyme Q10 are strong antioxidants that protect the gums and periodontium against free radicals that cause cell damage. Coenzyme Q10 is additionally necessary for the production of cellular energy in the mitochondria-it is a component of the respiratory chain in the cellular mitochondria. Its deficiency adversely affects the energy processes in the cell. It has been known for many years to improve the health of the tissues in the mouth. It is a catalyst for metabolic processes that provide the body with energy for repair processes and eliminates free radicals in cells thanks to its strong antioxidant properties [113,114,115,116,117].

Minerals are also essential for life. Like most vitamins, the body cannot produce them itself, so they must be supplied with food [118]. They are absorbed directly from food and are present in food in minimal amounts. There are micro- and macroelements. Micronutrients are those whose daily requirement is less than 100 mg, and macronutrients are elements that the body needs more than 100 mg a day. Some of these minerals are needed for efficient functioning in functional regulatory systems that influence each other, e.g., participate in the transmission of nerve signals. Others act and are part of hormones, others act as electrolytes and maintain adequate osmotic pressure in the blood. Contrary to vitamins, most of them are not sensitive to food processing methods, they are not damaged by heat and air. When cooking with large amounts of water for a long time, some of the water is rinsed out and if the water is poured out, these minerals are lost [113,114,115,116,117,118].

The abovementioned division is important if the proper diet is used, which helps to improve the immune system and defend against pathogenic bacteria causing gingivitis, which can turn into periodontitis. The proper components of the diet are listed below.

#### 2.1.1. Collagen 

Plaque, which is the leading cause of gum disease, builds up daily on the surface of your teeth. If we do not remove it regularly during brushing, pathogenic bacteria attack the gums in which the inflammatory process is triggered, and then further, deeper periodontal tissues are destroyed [119]. The process of degradation of collagen fibers takes place in parallel, which are like scaffolding for the gums and constitute up to 65 percent their volumes [119]. This is due to the release of collagenases, a group of enzymes that break down the natural collagen in the gum. Loss of collagen is characteristic of the early stage of inflammation, the gums become less resistant to infection and damaged tissues regenerate less [120]. That is why collagen-containing foods are so important. Sources of collagen necessary for the health of the gums, but also joints, cartilage and intestines, are, among others, bone broth cooked with chicken, beef or pork bones. Likewise, fish or pork jelly and offal [120].

#### 2.1.2. Coenzyme Q10 

This powerful antioxidant is found throughout the body and is a key ingredient for the proper functioning of every cell in the body [121].-People with adequate levels of CoQ10 are less likely to develop gingivitis and periodontitis. In turn, its deficiencies are found in 60–90 percent. patients with periodontal disease [121]. This compound reduces gingivitis, slows down its progression and stimulates repair processes. Therefore, it is worth supplementing it or eating foods containing it, the source of which is chicken meat, e.g., legs-necessarily with the skin, which is rich in collagen. The vegetable with the highest content of coenzyme Q10 is fresh spinach [121].

#### 2.1.3. Catechins 

These chemical compounds from the group of polyphenols have become one of the most powerful antioxidants with anti-caries, anti-inflammatory and antibacterial properties. The studies were performed on rats [122,123]. We can find catechins, among others in legumes such as broad beans, beans, lentils, cocoa, but their main source is unfermented teas. Research shows that green tea has a positive effect on overall oral health, while reducing the risk of gum disease because it contains as many as four catechins—epicatechin (EC), epigallocatechin (EGC), epicatechin gallate (ECG), and epigallocatechin gallate (EGCG)—which all inhibit the growth of bacteria responsible not only for caries but also periodontal disease, including *P. gingivalis, A. actinomycetemcomitans* and *P. intermedia* [124,125].

#### 2.1.4. Vitamin C

The link between bleeding gums and vitamin C levels was noticed over 30 years ago. This ingredient affects the process of collagen and connective tissue biosynthesis, it also supports the immune system in the fight against bacteria responsible for the development of periodontitis and improves the regeneration of the gums. Vitamin C deficiency, on the other hand, may increase the risk of gingival bleeding and promote inflammation [126]. Recent research from The University of Birmingham Periodontal Research Group (Birmingham, UK) confirms that gum bleeding tendency is associated with low plasma vitamin C levels. What’s more, scientists have found that increasing its daily intake helps reduce these symptoms. In this case, it is worth following a varied diet or considering vitamin C supplementation at a dose of about 100–200 mg per day [126]. The greatest amount of vitamin C is found in unprocessed products such as kale, kiwi, broccoli, parsley, red and green peppers-a glass of the former contains over 300 percent. recommended daily consumption, and another 200 percent. People who are on a specialized diet, e.g., paleo, in which products with vitamin C but containing sugar (e.g., kiwi, oranges) should also be careful about the deficiency [127].

#### 2.1.5. β-Carotene

β-Carotene, or vitamin A provitamin, is another essential nutrient supporting the health of our gums. Studies have shown that eating foods containing β-carotene helps reduce inflammation and aid in treating gum disease [127]. The greatest amount of β- carotene is found not in carrots, as commonly believed, but rather in sweet potatoes. Sweet potatoes also contain large amounts of vitamin C, vitamin B_6_, and manganese.

#### 2.1.6. Omega-3 

Omega-3 fatty acids are known for their anti-inflammatory properties and they are also a way to improve immunity and strengthen the body in general. Based on available studies, it appears that higher docosahexaenoic acid (DHA) intake has a protective role, reducing the risk of periodontal disease [128,129]. Their sources are fatty fish such as salmon, mackerel, herring, as well as macadamia nuts, pistachios and sesame.

#### 2.1.7. Fungotherapy Beneficial for Periodontium 

It is also worth paying attention to the inconspicuous shiitake mushrooms, commonly used in the kitchen, but also in Far East medicine, because they contain lentinate, a polysaccharide that has antibacterial properties, and also increases the production of antibodies, interferon and interleukins that stimulate the immune system to defend itself against infections. The research showed their anti-inflammatory properties and reduction of inflammatory markers. Shiitake mushrooms are a natural source of B vitamins that contribute to overall oral health [129,130,131,132,133]. Although it is difficult to talk about a diet against periodontitis-the most chronic gum disease, the mechanism of which is much more complex, dentists increasingly sensitize patients to the beneficial effects of a balanced menu that supports patients. Vitamin and mineral deficiency significantly increases the risk of periodontal disease [132,133]. Therefore, regardless of whether we are already under the care of a periodontist and we are struggling with gum disease, or we want to prevent such problems in the future, by including products rich in these nutrients in our diet, we support the health of our gums [132]. To reduce the growth of pathogenic bacteria, it is also worth eliminating sugar from your diet, which is a nutrient for them, and refined carbohydrates, such as white bread, rice or pasta. Let’s also take care of the correct oral microbiome by introducing probiotics into the diet that support the balance of the natural bacterial flora, inhibit the development of gingivitis and the accumulation of dental plaque. A natural source of probiotic bacteria is, for example, kefir, kimchi, silage, e.g., cucumbers or cabbage, popularly known as superfoods [132].

### 2.2. Diet and Oral Health

The outside world with its flora and fauna is essential to human life. It is used to obtain food that provides many stimuli to enjoy, and it is also the fuel needed to live/be-for the development of the physical side of a person-his body, metabolic processes and life energy. The oral cavity is a special place in the human body. It is there that you can taste the food that you can enjoy more or less, it is where the first digestive processes take place, which give rise to further processes necessary for the functioning of the macroorganism. It is there that pathological conditions arise, which in many situations generate systemic health problems. Pathological conditions in the oral cavity are the result of ineffective removal of food residues after meals and the active activity of the bacterial microflora inhabiting there. All living micro- and macro-organisms need food to survive, and the oral micro-world needs food scraps to thrive. Its development is unfortunately unfavorable for the macroorganism—its host. It damages the dental apparatus, as a result of which the host cannot use it effectively, which causes a cascade of adverse events. First, food that is not “worked out” properly in the mouth will not be able to be fully used by the body as it passes through the gastrointestinal tract [15]. Each section of the digestive tract has its own tasks. In the mouth, food is broken down by the teeth, softened by saliva, and thanks to amylase (an enzyme contained in saliva) it starts the process of digesting sugars. Amylase breaks down starch and other polysaccharides into simple sugars. The next stage of digestion takes place in the stomach and further in the subsequent sections of the digestive tract, until it is saturated with what the body will be able to obtain from it, and its remnants will be excreted outside. The human body is an very specific organism, equipped with defense mechanisms (specific and non-specific) that are triggered in emergency situations. These mechanisms work effectively when the macro-organism is healthy, nourishes properly and regularly uses proper oral hygiene. The antibodies make it difficult for bacteria to colonize the tissues and block their metabolism, but there are microbes that can destroy them. Other defense mechanisms are the continuity of the enamel and mucosa (constituting a natural barrier and protection of tissues against the penetration of microorganisms), exfoliation of the epithelium and bacteria deposited on them, the presence of bacterial flora (preventing the deposition of bacteria), movements of the tongue, cheeks and saliva (cleaning the surface of the teeth). As we know, saliva hinders the colonization of microorganisms and contains bactericidal substances (lysozyme, lactoferrin, histatin, staterin, apolactoferrin, bacteriocins and the sialoperoxidase system). In a situation where the body is subjected to ultraviolet radiation, ionizing radiation, ultrasonic waves, xenobiotics (along with food), oxygen consumption (which in 5% undergoes an unfavorable transformation, resulting in free radicals), oxidative stress will arise [119,120,121,122,123,124,125,126,127,128,129,130,131,132,133,134,135,136,137,138,139,140,141,142,143,144,145,146,147,148,149,150,151,152]. Oxidative stress is the result of disturbed homeostasis in the body, which can lead to irreversible changes. Low and high levels of oxidative stress mobilize cellular antioxidant mechanisms and stimulate the inflammatory response of cells, but very high levels of oxidative stress contribute to cell death (apoptosis and necrosis). A positive thing in this situation is the fact that maintaining an appropriate level of oxidative stress significantly influences the treatment of many inflammatory diseases [82,83,84,85,86,87,88,89,90,91,92,93,94,95,96,97,98,99,100,101,102,103,104,105,106,107,108,109,110,111,112,113,114,115,116,117,118,119,120,121,122,123,124,125,126,127,128,129,130,131,132,133,134,135,136,137,138,139,140,141,142,143,144,145,146,147,148,149,150,151,152].

We propose the use of “protocols” of four diets containing individual nutrients that should reduce inflammation and the formation of pathogenic bacteria in the mouth:Diet F including meals, containing proteins, carbohydrates-sugars, fats, vegetables.Diet B. Mainly targeted at protein products. You can eat other foods as well, but end each meal with a sugar-free protein product such as kefir, yoghurt, cheese, etc.Diet W. Mainly oriented towards vegetables and other foods can also be eaten, but each meal should end with vegetables, such as radish, watercress, kale, broccoli, kohlrabi, etc.Diet T. Mainly targeted at foods containing Omega-3 fatty acids, can also eat other foods, but each meal should be finished with food containing Omega-3 fats, e.g., fish-especially salmon, herring, mackerel, sardines, seafood, sushi, rapeseed oil, linseed, soybean oil, soy products, nuts, almonds, pumpkin seeds.

One should eat meals consisting mainly of products from the recommended diet (diet types F, B, W, T), and at least finish each meal with a product from the recommended diet (own research).

Due to the fact that one of the non-specific defense mechanisms is the continuity of the enamel, it is necessary to look at the phenomenon that leads to its destruction. This process is called caries. Dental caries is a disease manifested by the demineralization of the hard tissues of the tooth-enamel and dentin (its inorganic part) and then by the proteolytic decay of the pulp (organic part), if left untreated, it usually leads to the death of the tooth and, unfortunately, quite often to its loss [49,50,51,52]. For this process to occur, four basic factors are needed, i.e., acid-forming bacteria that inhabit the bacterial plaque, the substrate, i.e., mainly disaccharides transformed by bacteria into acids, time and susceptibility of tooth tissues (genetic and environmental conditions). Given these factors, bacteria, mainly *Streptoccocus mutans* and *Streptococcus sobrinus*, need carbohydrates. These bacteria metabolize carbohydrates to produce acids that dissolve the hard tissues of the tooth. In the carious process, the amount of saliva produced in the oral cavity is important, with its deficiency, caries progresses very quickly. The acid formed as a result of carbohydrate metabolism works for half an hour, then saliva neutralizes it, then the enamel is remineralized with calcium and phosphorus, which are released from saliva (in such a situation, caries may develop for years or not at all) [129,130,131,132,133,134,135,136,137].

The process of restoring damaged tooth tissues only takes place when the pH in the oral cavity is maintained at <5.5 for a long time. Unfortunately, carbohydrates rapidly lower the saliva pH below 5.5 and initiate tissue demineralization. As the frequency of carbohydrate consumption increases, the time the pH remains below the critical level increases, and the lower the higher the sugar concentration [125,137,138,139,140,141,142,143]. 

Not all carbohydrates work the same way. The most dangerous are simple sugars (glucose and fructose) and disaccharides (sucrose and maltose). They convert into harmful acids the fastest. Galactose and maltose show less cariogenic properties. In addition, dairy products containing these sugars, rich in calcium, phosphorus and protein-neutralize the resulting acids and contribute to the restoration of enamel. Complex carbohydrates, i.e., starch (rice, potatoes, flour), are less cariogenic, but they can also initiate the cariogenic process. It has been tested and proven that products containing sucrose and starch (e.g., sweet rolls) are much more cariogenic than those containing only sucrose. In addition to products containing acidic sugars (e.g., sweetened carbonated drinks or fruit juices), the pH value of many fruit juices and sweetened carbonated drinks is in the range of 2.1–4.46, and in the etiopathogenesis of caries it is important how often we eat caries (caries), what is the consistency of the food we eat [125,137,138,139,140,141,142,143]. The time of neutralization of the acidic environment by saliva after consuming sucrose is about 40 min, and in the case of starch products with the addition of sucrose-up to 2 h. Foods with a sticky texture stay on the tooth surface for a long time. Frequent consumption of sugar-containing foods, especially soft and sticky ones, helps to maintain the acidity of the plaque and to the demineralization of the enamel, i.e., the loss of minerals. Unfortunately, it is also favored by eating foods or liquids containing sugars immediately before bedtime, as the amount of saliva produced during sleep is significantly reduced. Eating sugar-containing products with the main meal is less harmful than between meals [144,145].

In some situations, carbohydrates can be replaced with sweeteners, the so-called sweeteners. Sweeteners are a good alternative to sugar, which, apart from empty calories, does not add any nutritional value to the body. Sweeteners have different properties [144,145].

Sweeteners are divided into three groups of compounds: natural sugars, polyols (so-called semi-synthetic fillers) and intense sweeteners (their small amount multiplies the sweetness of sucrose). Natural sugars are mono and disaccharides, i.e., glucose, fructose, sucrose and sugar syrups. The second group includes sorbitol, lactitol, xylitol-they affect the texture and, for example, maintain moisture and are less sweet than sucrose (40% less calories). The third includes: acesulfame K, aspartame, saccharin, cyclamates, sucralose, steviol glycosides. Substitutes are agents belonging to the second and third groups. Intensive sweeteners in the body are not metabolized, they are weaker than sucrose and are treated as so-called sweeteners. non-nutrients-they do not provide energy (i.e., calories). Their big advantage is the fact that they do not ferment in the mouth and therefore do not cause tooth decay. For this reason, polyols are used in the production of mouthwashes, toothpastes and chewing gums. Polyols used in larger amounts may have a laxative effect. Agents that have been approved for use in food have been thoroughly tested in terms of their impact on human health. In the European Union countries, sweeteners are approved for use: acesulfame K, aspartame, cyclamic acid, isomalt, sorbitol, mannitol, maltitol, thaumatin, neohesperidin, lactitol and xylitol. In Poland, aspartame and acesulfame K have been approved for use. Intensive sweeteners, weaker than sucrose, should meet certain conditions, i.e., have sweetness such as sucrose, be safe for health, be cheap and functional, demonstrate resistance to digestive processes in the digestive tract, not they should provide energy and cause tooth decay, allergies and diarrhea. Intensive sweeteners are divided into natural (stevidosis, glycyrhizin, thaumatin, monellin, pentadine, curculin-in Poland only thaumatin is approved for use in food) and artificial (saccharin, aspartame, acesulfame K, cyclamates, alitama, sucralose) [125,144,145,146,147,148,149,150,151,152,153,154,155,156,157,158,159,160,161,162,163,164,165,166,167,168,169,170,171,172,173,174,175,176,177,178,179,180].

The fourth factor complementing the possibility of the formation and development of caries is the individual tooth susceptibility. When speaking of susceptibility, we take into account the hardness, anatomical structure of the teeth, their location in the arch, their quantitative composition and the degree of enamel and dentin mineralization-they determine the strength of the tooth exposed to harmful factors-mainly acids. Many people believe that we are born with sensitivity and we have no control over it, but it can be influenced by the mother who feeds the baby during pregnancy [85,86,87,98].

In the prenatal period, the susceptibility of human teeth is formed. The baby’s mother’s nutrition and oral health are very important. It would be great if a woman planning pregnancy prepared herself properly for this process, i.e., had clean teeth, healthy periodontium, properly maintained oral hygiene and properly nourished herself. During pregnancy and lactation, the woman’s body needs not only energy, but also nutrients-minerals, i.e., calcium, phosphorus, magnesium, iron, zinc, copper, iodine, selenium and vitamins: A, B1, B2, niacin, choline, pantothenic acid. B6, B12, C, E, and folic acid. A balanced diet of the future mother determines the proper formation and mineralization of tooth tissues. In the embryonic period, tooth buds begin to form, and their further development and the beginning of mineralization continue throughout pregnancy. Milk teeth begin to form around 6 weeks after conception from the cells of the fetal mouth. The process of tooth mineralization begins in the bell phase of tooth development, or around the 4th month of pregnancy, during the formation of dentine on which the enamel is deposited, and continues until puberty. Insufficient degree of tooth mineralization (enamel and dentin) is a risk factor for the development of early childhood caries [113,114,115,116,117,118]. Proper mineralization of the tooth tissues in this period determines the resistance to caries, depends mainly on the supply of mineral salts-calcium, phosphorus, fluorine, magnesium, molybdenum, manganese and vitamins, especially A, C and D. The influence of these components is also of great importance for the health of the cavity. Oral during the development of tooth buds. Vitamin A and D deficiencies as well as protein and energy deficiencies during pregnancy are a risk factor for the development of enamel hypoplasia-enamel underdevelopment in the form of hypoplastic defects, tooth morphogenesis disorders, delayed eruption, disorders of differentiation and odontogenic functions, which may be manifested by atypical dentin formation and disorders of mineralization of the teeth and the loss of salivary glands in the child, which in turn can lead to a reduction in the buffering capacity of saliva and thus make the teeth more susceptible to decay [113,114,115,116,117,118]. Vitamin deficiency occurs in the case of malnutrition caused by a lack of fats, carotenes, elimination of dairy products, fresh fruit and vegetables, in the case of digestive disorders and fat absorption, and liver failure. Lack of adequate vitamin A also causes early tooth decay. Keep in mind that retinoids (vitamin A derivatives) are teratogenic, while carotene—a form of vitamin A found in fruits and vegetables—is not harmful. Vitamin D_3_ is an essential catalyst for calcium and phosphate metabolism. Together with parathyroid hormone and calcitonin, it is responsible for the mineralization and resorption of bone tissue, as well as the absorption, use and regulation of phosphate and calcium levels in the body. Its deficiency in the oral cavity is manifested by a reduction in the size of the dental arches, deformation of the jaw bone, alveolar process (which contributes to the formation of occlusive disorders), abnormalities in tooth eruption, and even tooth retention [113,114,115,116,117,118]. Another consequence of its absence may be disorders in the functioning of ameloblasts with insufficient mineralization of enamel, dentin and root cement, delayed eruption and reduction in the size of molars. Primary teeth hypomineralization can occur not only as a result of vitamin D deficiency, but also calcium and phosphate deficiency (which also increases the risk of caries in early childhood). It is also not recommended to use an excess of vitamin D, it may lead to changes in the structure of the tooth, e.g., creating a thinner enamel layer. This can be manifested by a violation of the integrity of the tooth tissues and a delayed eruption time. The abnormalities in the formation of tooth tissue in the case of calcium, phosphate and vitamin D deficiency during pregnancy are irreversible and can affect both deciduous and permanent teeth. Vitamin C has a significant impact on the development of dentin, it is necessary for the proper integrity and functioning of odontoblasts, fibroblasts and chondroblasts, it is a substance necessary in the process of collagen synthesis. Collagen is the backbone of the organic matrix in the deposition of phosphate and calcium crystals and bone mineralization [85,113,114,115,116,117,118] (Figure 4).

Vitamin C deficiency during odontogenesis causes odontoblast atrophy, irregular, reduced dentin deposition, and disrupts tooth growth. It causes fragility of the vessels in the pulp, it also leads to a disturbed function of odontoblasts, which results in underdevelopment of the dentin. Long-term deficiency of this vitamin causes scurvy, which causes swelling, bleeding gums and loss of teeth. Vitamin C works in concert with vitamin A to promote tooth development and mineralization. It should be noted that, according to some authors, in the 4th month of pregnancy (according to others, already by the 8th week), the fetal taste receptors begin to develop. A mother’s consumption of large amounts of sweets, occuring during this period, may increase the child’s tendency to sweet foods in the future. Tooth compliance is of particular importance in children with tooth growth and mineralization, but also in the elderly whose teeth have already undergone some damage [85,113,114,115,116,117,118].

The eruption of primary teeth begins in the second half of the child’s life. When a child is fed (naturally or artificially), too often and excessively long, before going to bed (during the day or at night, when there is less salivation) and if the child’s teeth are not cleaned, the enamel becomes demineralized and, consequently, caries. After the child reaches 17 weeks of age (up to 26 weeks of age), it is recommended to introduce into the daily menu next to dairy foods, gradually supplementary non-dairy foods, and at the age of 9–24 months-snacks such as vegetables, fruit and bread [153,154,155,156,157,158,159,160,161,162,163,164,165,166]. After the age of two, it is beneficial for your child to eat 4–5 meals a day, avoid extra snacks, and leave at least 2 h between meals to allow saliva to neutralize acids and repair enamel. It should be remembered that the teeth are used to bite and in the first two years of life. Along with the eruption of subsequent milk teeth, the child should develop this skill, because thanks to this he will have a chance to develop his stomatognathic system (prevention of malocclusion and, consequently, periodontal diseases over time) and will gradually reduce the suckling reflex. Therefore, after the age of 6 months, introduce less crushed foods and do not give food and liquids through the bottle with a teat [166]. Biting on hard, high-fiber foods also promotes salivary secretion (stimulated salivary gland function). In the case of protein and energy malnutrition, iron, zinc and vitamin A deficiency and dehydration, the secretion of saliva and its composition change, the concentration of total protein, α-amylase, lysozyme and IgA in saliva decreases. Therefore, it is recommended to eat products containing fiber, especially hard, raw vegetables and fruits (water contained in fruits and vegetables and stimulation of saliva secretion reduces the effect of sugars contained in them) and grains and whole grains rich in potassium, magnesium, selenium, zinc, vitamins. from group B (B_1_, B_2_, B_6_, PP, folates) and vitamin E [113,114,115,116,117,118]. Fiber increases the volume of food, facilitates the removal of food debris and toxins, slows down the absorption of glucose, and thus reduces a one-time glucose release. As well as dairy products (milk, hard cheeses, which contain calcium and phosphorus necessary for remineralization), thanks to which lipids form a protective coating on the surface of the teeth against acids and stimulate salivation. Consuming cheese (especially hard cheese) should end the meal as it quickly increases the pH of saliva. Cheese casein phosphopeptide has been shown to form a complex with calcium phosphate (CPP-CP), increasing the pH of plaque and acting as an ion reservoir for enamel remineralization. In addition, products rich in protein (meat, poultry, fish, eggs), phosphorus and proteins containing arginine (e.g., sunflower seeds, pumpkin seeds, squash, nuts, coconut, beans, soybean, watermelon and tuna) are recommended-they have the ability to quickly increase pH. A significant reduction in the supply of sugar was recommended [114,132,144].

Due to the cellular structure of food and lower availability of bacteria, internal sugars (e.g., sugars contained in apple eaten raw, are less cariogenic than sugars in apple juice or baked apple) and external milk sugars (accessible to bacteria, non-cariogenic thanks to protective ingredients found in milk). In the first two years of life, it is not recommended to add sugar to meals and snacks (including natural sugars in the form of fruit syrups or honey) and salt. This has a positive effect on general health and dentition and enables the development of taste preferences that are beneficial to health. The World Health Organization (WHO) recommends limiting the consumption of free sugars to less than 10% E (with an energy consumption of 1900 kcal, this corresponds to about 48 g, i.e., about 10 teaspoons per day). According to the latest reports, it is beneficial for general health and dentition to reduce sugars to <5% E. When choosing food products, one should take into account the sugar content and the probable food retention time on the tooth surface (consistency and stickiness). Take cocoa for example-it has anti-cariogenic potential, but chocolate consumption lowers the pH to a level critical for enamel. Dark chocolate is less cariogenic than milk chocolate, but the cariogenicity of chocolate is increased by the addition of raisins and fruit. The duration of action of acids is longer after eating foods containing sucrose and sticky starch, causing prolonged contact with the tooth surface (eating lollipops and lozenges has a similar effect). Acidic products, sweetened and carbonated drinks should be avoided-they have a particularly detrimental effect on the condition of the teeth. For example, orangeade and cola drinks contain about 10 g of sugar per 100 mL of drink, lemonade about 6 g. This means that one glass of this type of drink is the dose corresponding to 3–5 teaspoons of sugar. A 355 mL can of sweetened soda contains up to 40 g (about 8 teaspoons) of sugar (WHO) [181]. Carbonated drinks additionally contain acids—mainly citric, apple, carbonic, orthophosphoric acid, and as an antioxidant-ascorbic acid—contributing additionally to the erosion of the enamel. Enamel demineralization begins when the pH is lowered to 5.5. It is recommended to drink low-mineralized, low-sodium, low-sulfate water in order to limit the drinking of juices and sweetened beverages as well as flavored waters [166]. In the current situation of the global COVID-19 pandemic, maintaining the principles of a proper diet with the above ingredients described above is essential for our overall health, including periodontal diseases which, if left untreated, can lead to periodontitis. The literature data based on the meta-analysis shows that patients with periodontitis diagnosed with COVID-19 were found to have over nine times higher risk of death than those without advanced periodontal disease [182]. Moreover, the risk of hospitalization in the intensive care unit is 3.5 times higher for them and 4.5 times the risk of having to use a respirator than for people with healthy gums [182]. A large-scale case-control study has been conducted around the world that used electronic medical records, including dental panoramic data. Measurement of inflammatory markers turned out to be higher in people undergoing COVID-19 and diagnosed with periodontal disease, which is associated with a higher risk of complications during the inflammatory process. The results of the study indicate that inflammation in the mouth paves the way for the coronavirus to attack more aggressively. Therefore, dental care should be part of recommendations that reduce the risk of severe COVID-19 [183]. The study included 568 patients who were diagnosed with COVID-19 between February and July 2020. Of these patients, 40 had complications that resulted in their admission to the intensive care unit, necessity to use a ventilator, and some of them died. The second study group-nearly as numerous-consisted of 528 people who had COVID-19 more mildly at home. Information has been gathered on gum disease and other factors that may be associated with complications of COVID-19, including body mass index (BMI), smoking, asthma, heart disease, diabetes and high blood pressure. Data on blood levels of chemicals related to inflammation in the body were also obtained. In the first group of COVID-9 patients, as many as 258 patients (45%) had gum disease. After taking into account age, sex, BMI, smoking and other conditions, it was calculated that the risk of complications of COVID-19 in people with periodontitis is 3.67 times higher, the risk of admission to the ICU is higher 3.54 times, the need to use a respirator may occur 4.57 more often and 8.81 times more often death is compared with people without periodontal disease [182,183]. Researchers found that if a causal relationship between periodontal disease and a higher risk of complications in people with COVID-19 can be established, gum treatment could become an important part of COVID-19 therapy in periodontal patients [182,183]. In addition, oral bacteria in parodontosis patients can infect the lungs, especially in ventilator users [184,185]. This may aggravate COVID-19 patients and increase the risk of death. Therefore, hospital staff should identify COVID-19 patients with periodontal disease and use oral antiseptics to limit the transmission of bacteria. All the more so as the relationship between periodontitis and lung diseases-asthma, pneumonia and chronic obstructive pulmonary disease-is well known [186,187,188]. Periodontal tissue inflammation is modulated by the host’s response to plaque build-up in the gingival fissure or pocket and destroys the tooth suspension. The gingival pocket, also known as the tooth pocket, is a gingival groove located near the neck of the tooth, between the gingival margin and its epithelial attachment. Its depth is determined by a special periodontal probe. It should not exceed 2–3 mm. If it becomes larger, it is considered a pathological phenomenon and indicates an ongoing lesion [186,187,188,189,190,191,192,193,194,195,196,197,198,199]. Then it begins to be accompanied by persistent, unbearable pain discomfort. The most common reasons leading to the formation of a gingival pocket at the tooth are, first of all, improper oral hygiene [200]. It leads to the development of a number of periodontal diseases (both aggressive and chronic) that significantly affect the condition of the gums [200,201,202]. The relationship between the systemic analysis of oxidative stress and the pathogenesis of the periodontal pocket, which is a pathognomonic change in periodontal disease, should also be emphasized. The control of the inflammatory network of the pockets, the analysis of the system for capturing oxidative molecules from the pockets and healthy gums, or at most gingival-root fluid exudation, (the role of the Human Gingival Crevicular Fluids-GCF protein) is very important in the pathogenesis and therapy of periodontal diseases (e.g., the mechanisms of their formation) [186]. Traditional periodontal diagnostics is based almost exclusively on clinical and radiological evaluation. It is now suggested that the local response of the body to periodontal disease could be evaluated by analyzing gingival pocket fluid (GCF) [186,203,204]. The study of humoral factors present in GCF can be of great diagnostic value, as GCF is the source of many factors involved in the immune-inflammatory reaction. The presence and appropriate concentrations of humoral factors in GCF can be used in the assessment of periodontal disease activity and the results of the applied therapy [186,198,204]. From a clinical point of view, therapies in periodontology are aimed at preventing or controlling the development of periodontal pockets. We can include them among them; -Analysis of the inflammatory proteomic and/or cytokine network of the periodontal pocket and the role of the GCF protein [186,198] and the relationship between systemic inflammation and GCF [186,198,203] and the relationship between GCF and periodontal pocket are extremely important [186,198,203,204]. According to Armitage [205,206], several dozen components of GCF can be used as potential diagnostic markers of periodontitis. They can be divided into three main groups: (1) inflammatory mediators and factors modulating the immune response, (2) host enzymes. Mediators as biomarkers and their inhibitors, (3) by-products of tissue destruction. The assessment of the level of selected cytokines and chemokines in GCF may be of particular importance in the diagnosis of periodontal diseases. Typical oxidative stress biomarkers-such as: resistins, adiponectin, TNF-α, leptin, IL-6, IL-8 and IL-1β have been detected in saliva and gingival fracture fluid (GCF) during chronic periodontitis (CP) progression [207,208]. Based on the data from the meta-analysis, an association was found between the biomarkers of oxidative stress and the progression of the gingival fluid [209,210,211,212,213,214]. It should be emphasized that macrophages and fibroblasts also accumulate in the inflamed periodontium. These cells are the source of numerous pro-inflammatory cytokines, mainly Il-1, Il-6 and TNF, as well as metalloproteinases (MMPs) and proteolytic enzymes [215,216]. However, many authors indicate that the measurement of the concentration of other humoral factors involved in regulating the course of immunological and inflammatory processes may also be extremely useful and helpful in the diagnosis and prognosis of periodontal diseases. Inhibition of bacterial adhesion and epithelial colonization is important from the point of view of immune mechanisms and is conditioned by numerous mechanisms, including primarily the flow of gingival pocket fluid (GCF) and its components (antibodies, proteases, complement components, antibacterial saliva components) [217]. In the initial phase of periodontitis, there is an increased flow of GCF and accumulation of neutrophils with concomitant loss of connective tissue. GCF is the blood plasma filtrate that contains a large amount of complement proteins. Activation of the complement system in an alternative way in the gingival pocket leads to the accumulation of the C3a and C5a components. Both proteins are anaphylatoxins which trigger the release of histamine from tissue-resident mast cells [218]. Histamine increases vascular permeability and causes swelling of the gums. At the same time, the components of the filler pathway are activated, as well as factors of bacterial origin. Developed habits are also important stimulants in the form of compulsive smoking of tobacco products, as well as an improper diet rich in carbohydrates [218]. There is a clear link between poor oral hygiene and gum health. Neglect of care promotes the formation of plaque and tartar, which gradually accumulates in the gingival groove. Then, microorganisms colonize this area in large numbers, which worsens the condition of the gums [198]. Pathological tooth pockets can also occur in the event of pulpitis. If the inflammation affects the periodontal tissues in the immediate vicinity of the affected tooth, problems may arise related to the deepening of the gingival pocket. At an advanced stage, bone defects and sometimes also purulent exudate from the dental pocket may occur. It is best to treat gingival pockets under the supervision of a periodontist who specializes in the treatment of periodontal diseases [198,199]. Normal proceedings are multi-stage. It is used to eliminate pain in the gingival pocket and calm the inflammatory process. The most important element of dealing with the problems of gingival pockets is, above all, prophylaxis, which includes developing proper hygiene habits. Regular tooth brushing with the use of appropriately selected accessories-a toothbrush, toothpaste, mouthwashes and dental floss-increases the chances of maintaining healthy teeth and periodontal tissues [186,201,202,203,204,205,206,207,208,209,210,211,212,213,214,215,216,217,218].

An extremely important and important aspect of the theses reported in the work that should be emphasized are potential clinical applications (implications) trying to answer the question; how traditional diets can influence oxidative stress and bacterial biofilm-induced inflammation in COVID19 patients-in relation to inflammation in mast cells. Since MCs are producers of histamine in inflammatory reactions, this active amine, by increasing the production of IL-1, can increase the inflammatory process in the lungs of SARS-CoV-2 infected. In the work of Conti et al. [219], the role of histamine released by MC was proposed for the first time, which in combination with IL-1 may increase pneumonia caused by SARS-CoV-2 viral infection. Histamine and IL-1 are involved in the pathogenesis of the lung inflammatory reaction after the activation of immune cells by pathogenic microorganisms belonging to all known and analyzed bacterial complexes. IL-1 in combination with histamine can cause a strong increase in IL-1 levels, leading to severe inflammation in the tissue. Furthermore, histamine enhances IL-1 induced IL-6 gene expression and protein synthesis via H 2 receptors in peripheral monocytes. Literature data show that histamine alone has no effect on the production of IL-1 [219]. Traditional diet rich in nutrients containing lactoferrin and probiotic strains such as L. salivarius can effectively block the multiplication of SARS-CoV-2 virus by activating innate immune and structural cells of the bronchial epithelium and endothelial cells, reducing pro-inflammatory cytokines (own research) and inducing differentiation MC mast cells. An important aspect is also the reduction of immunity in patients undergoing implant placement [220,221,222]. The study by Torrejon-Moja et al. [220] assessed the survival of implantoprosthetic rehabilitation in controlled HIV-infected patients with good oral hygiene. Each patient received at least one dental implant. After 90 days in the upper jaw and 60 days in the lower jaw, an appropriate prosthesis was made, the primary endpoint of which was prosthetic failure, implant damage, changes in the marginal bone level around the implant (MBLC) and biological complications (peri-implantitis, pus, pain, paresthesia). The study has demonstrated in its limitations that in a well-controlled HIV patient population, implant rehabilitation may be an appropriate option in developing an appropriate infection control protocol [220]. An important aspect is also the reduction of immunity in patients undergoing implant placement [220,221,222]. HIV-infected patients with edentulousness who required prosthetic restoration on one or both jaws were also included in the study. Each patient received at least one permanent full arch prosthesis [221,222]. Marginal bone loss, implant and prosthesis failure, biological and mechanical complications, and serological levels (Cell differentiation antigen-CD4 cell count, CD4/CD8 ratio and HIV viral load) were recorded up to 7 years of follow-up. Within the limitations of this prospective 7-year longitudinal study, HIV infected patients with a stable immune system may be treated under the “all-four protocol”. Attention should also be paid to the use of porcine bone biomaterials for transplant to minimize alveolar bone collapse after tooth extraction by histomorphometric and in vivo osteoblast-specific gene expression profiling for Runx2, osteopontin, osteoprotegerin, type I collagen and alkaline phosphatase real-time reverse transcriptase polymerase chain reaction [223]. In this study, histological examination and biomolecular evaluation confirmed good biocompatibility and high osteoconductivity of porcine xenogeneous bone in alveolar bone grafting [223]. Also the use of synthetic magnesium enriched hydroxyapatite (MHA) with porcine bone (PB) grafts in fresh sockets by histological and histomorphometric analyses. Histological examinations showed the absence of inflammatory cells, bone formation in all treated areas and the presence of biomaterial and connective tissue particles [223]. Nalzey also mentioned the technique of sinus lift [224] which may be a complementary test (complementary to the studies described earlier [221,222,223]. In the sinus lift technique, an important element is the evaluation of the clinical results and radiographic data of the percutaneous sinus lift (TSFE) of the residual alveolar bone ≤3 mm. 46 patients with edentulousness in one or both of the posterior segments of the jaw were enrolled in the study. The residual ridge of the alveolar ridge was measured. TSFE was performed without bone graft. Three months after the first surgery, 66 implants were placed without material for transplantation. The preoperative distance from the alveolar crest to the bottom of the maxillary sinus and the amount of new X-rays between the sinus floor and the alveolar crest were measured from the mesial and distal surfaces of each implant surface. The results of this retrospective clinical trial confirmed the reliability of the TSFE procedure and the maintenance of bone levels over time without transplant surgery [224,225]. Also an excellent tool for this type of technique is the use of an osteotome for vertical bone augmentation and local elevation of the local sinus floor (LMSF) in fresh molars over a period of 13 years of follow-up, with minimal surgical trauma, as an appropriate procedure to increase the vertical size of the bone available for implantation [225]. The results of the conducted studies showed that the LMSF procedure in fresh molars allowed to increase the dimensions of the resorbed posterior alveolar bone of the maxilla both vertically and horizontally, with the effectiveness of 100% osseointegration of the implant over time. Therefore, the ingredients in our diet are very important, which “support and help” based on the analysis of the above techniques in maintaining proper homeostasis and oral microbiota.

## 3. Conclusions

Periodontal diseases are still one of the major health problems in Poland. Statistics show that out of 10 people, nine have periodontal problems [134]. The development of the disease is a consequence of an imbalance between the potentially pathological bacteria found in the oral cavity (supra- and subgingival plaque) and the host’s immune response. In the oral cavity, homeostasis can be modified by a number of constant (congenital) and variable (acquired) factors that are risk factors for developing disease. The most important congenital risk factors include age, genotypes (genetic factors), gender, and race. The second group includes variable determinants, i.e., improper oral hygiene, local factors of plaque accumulation (gingival areas, inadequately placed fillings, tooth crowding, cervical areas of crowns, bridges), unfavorable composition of bacterial biofilm present on the teeth, use of tobacco (smoking pipes, cigarettes, cigars, the use of non-flammable tobacco-chewing snuff), nutritional deficiencies (lack of vitamin C, calcium), diseases (e.g., diabetes, alcoholism, osteoporosis) and long-term exposure to stressors [113,114,115,116,117,118,153,154,155,156,157,158,159,160,161,162,163,164,165,166],

Considering the cariogenic factor, the diet should be rich in calcium, phosphorus, fluorine and vitamin D products, as well as nutrients involved in the growth and mineralization of teeth. Based on the quoted literature data, recommendations can be made regarding the change of eating habits in the diet to reduce the risk of oxidative stress and inflammation that may contribute to the development of pathogenic microflora in the oral cavity. One should definitely limit the consumption of carbohydrates, especially those with a viscous consistency, sweetened drinks, sour drinks, carbonated drinks and juices. People with healthy temporomandibular joints may be advised to chew sugar-free chewing gum after a meal (but for about 5–10 min) when they cannot brush their teeth. Hygiene that is appropriate to the age and situation in the mouth is very important. Considering the erosive factors, in addition to the above, avoid drinking beverages lowering the pH below 4.5, it is recommended to drink them through a straw. It is advisable to end your meals with foods that neutralize the pH of the oral cavity (e.g., cheese, dairy products, milk). It is recommended to drink water, and if you drink acidic drinks, you should sweeten them a little before drinking (then the pH becomes neutral), do not drink between meals and in the evening. Allow 1–2 h breaks between meals to facilitate the remineralization of hard tissues. It is recommended not to brush your teeth immediately after consuming acidic foods or liquids so as not to exacerbate erosive changes. By following the above recommendations, we can hope to avoid serious inflammation, even leading to tooth loss, through chronic inflammation of the tissues of the attachment apparatus. We hope that our review article will be a compendium of knowledge on the recommendations for choosing the right diet for daily oral hygiene. This study highlights a different relationship between gum disease and systemic health, and confirms the need for continuous, systematic dental care for periodontal-prone people and for strong periodontal prophylaxis for the entire population. Periodontal diseases affect up to 50 percent of all adults around the world and they are recognized by various dental associations and federations as a disease of civilization.

## Figures and Tables

**Figure 1 materials-14-01444-f001:**
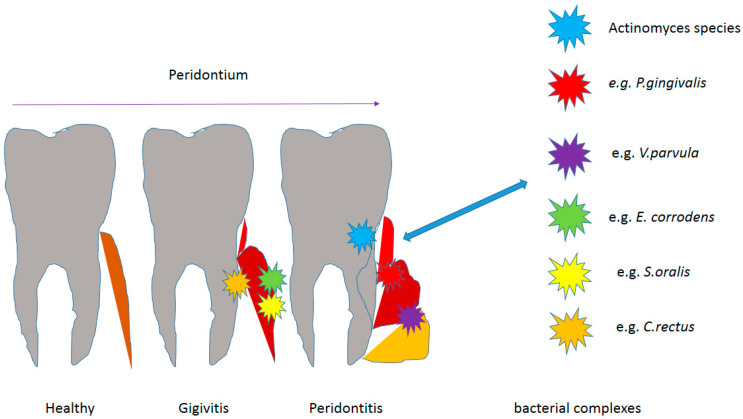
The process of periodontal disease formation according to Socransky [179,180].

**Figure 2 materials-14-01444-f002:**
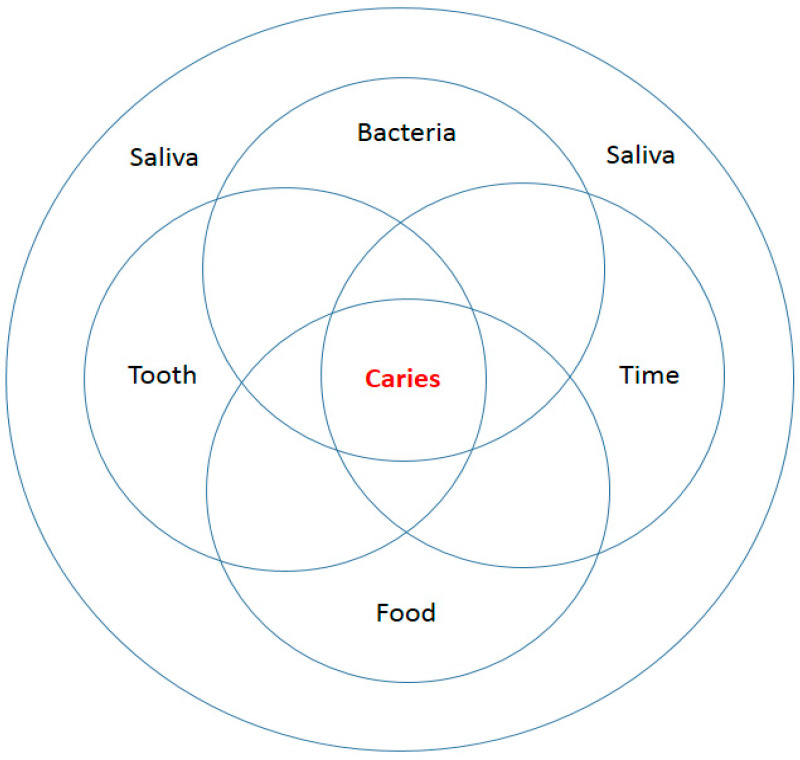
Caries-generating factors.

**Figure 3 materials-14-01444-f003:**
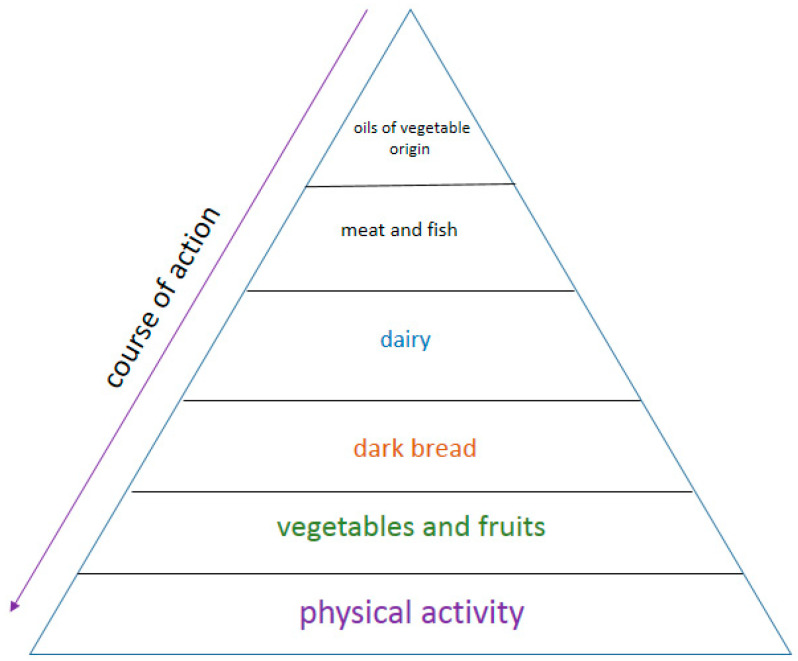
Food pyramid.

**Figure 4 materials-14-01444-f004:**
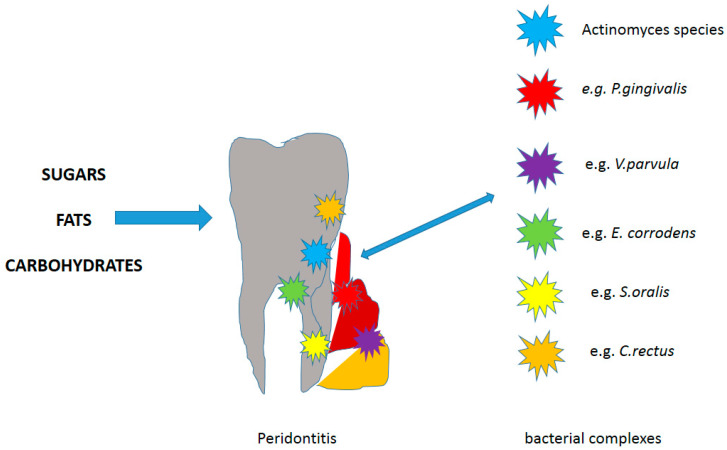
The role of nutrients in the body according to [169].

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
