# Peer review of "The Influence of Diet on Oxidative Stress and Inflammation Induced by Bacterial Biofilms in the Human Oral Cavity"

_materials, 2021, doi:10.3390/ma14061444_

Round 1

Reviewer 1 Report

Esteemed colleagues,

The topic is actual as diet and periodontal disease are a subject of interest in the current literature.

The paper is extensive and quite well organised.

I would  suggest to add a more detailed protocol of modifying the diet in such a way that will reduce periodontal risk.

Author Response

Thank you very much for the apt suggestions and comments of the reviewer, which contributed to the improvement of the substantive value of our manuscript. We kindly ask you to comment on our amendments positively.

Reviewer 1

Esteemed colleagues,

The topic is actual as diet and periodontal disease are a subject of interest in the current literature.

The paper is extensive and quite well organised.

I would  suggest to add a more detailed protocol of modifying the diet in such a way that will reduce periodontal risk.

In our new article on the influence of different diets on oral hygiene (and which is also currently in reviews), we have provided detailed information on the protocols for the use of individual types of four diets, the product content of which is given below in examples:

Diet F including traditional meals, containing proteins, carbohydrates - sugars, fats, vegetables.

Diet B. Mainly targeted at protein products. You can eat other foods as well, but end each meal with a sugar-free protein product such as kefir, yoghurt, cheese, etc.

Diet W. Mainly oriented towards vegetables and other foods can also be eaten, but each meal should end with vegetables, such as radish, watercress, kale, broccoli, kohlrabi, etc .

Diet T.Mainly targeted at foods containing Omega-3 fatty acids, can also eat other foods, but each meal should be finished with food containing Omega-3 fats, e.g. fish - especially salmon, herring, mackerel, sardines, seafood, sushi, rapeseed oil, linseed, soybean oil, soy products, nuts , almonds, pumpkin seeds.

Eat meals consisting mainly of products from the recommended diet (diet type F, B, W, T), and at least finish each meal with a product from the recommended diet

The abstract has been rebuilt as well as the summary chapter. The entire manuscript has been thoroughly reworked, descriptions and division of periodontal diseases have been added, and the key components of the diet, which are also the protocol for its use, have been included (page 15 and pages 13 and 14 in the manuscript), all corrections are marked in green.

The entire manuscript has been thoroughly reworked, descriptions and division of periodontal diseases have been added, and the key components of the diet, which are also the protocol for its use, have been included (page 15 and pages 13 and 14 in the manuscript), all corrections are marked in green.

Reviewer 2 Report

Dear Authors,

The article entitled "The influence of traditional diet on oxidative stress and inflammation induced by bacterial biofilm in oral cavity in mammalians" is an extensive and  narrative exposure  on caries, periodontal diseases and their causal microorganisms. 

There are some small corrections that I suggest only in order to be more adequate:

In the introduction in the description of the Buccal epithelium: The epithelium covers the entire surface of the mouth. should be changed with interior of the cheek

Page 13: Inflammation is a consequence of subsequent phenomena caused by the imbalance of the oral cavity bacterial flora. In my oppinion this is not only the cause for inflammation as it can be induced by autoimmune disorders and many others. So it should be Inflammation may be/appear as a consequence..

Page 13: exfoliative gingivitis is not the best choise in the context. It should be changed in desquamative gingivitis.

Page 18: Smoking cigarettes upsets the balance. Replace upsets with causes imbalance 

The content data of Figures 4 and 5 are also included in Figure 7. In my oppinion Figure 4 and 5 can be removed.

Page 24: The conclusion chapter does not reflect data presented in the manuscript. It should be mentioned from the Introduction: Periodontal diseases are still one of the major health problems in Poland. Statistics show that out of 10 people, 9 have periodontal problems. Citation is needed.

The article needs minor revision before publication.

Best regards!

Author Response

Thank you very much for the apt suggestions and comments of the reviewer, which contributed to the improvement of the substantive value of our manuscript. We kindly ask you to comment on our amendments positively.

Reviewer 2

Thank you very much for your pertinent comments, which contribute to improving the substantive quality of our work. All revisions in the manuscript are highlighted in green.

The article entitled "The influence of traditional diet on oxidative stress and inflammation induced by bacterial biofilm in oral cavity in mammalians" is an extensive and narrative exposure on caries, periodontal diseases and their causal microorganisms.

There are some small corrections that I suggest only in order to be more adequate:

In the introduction in the description of the Buccal epithelium: The epithelium covers the entire surface of the mouth. should be changed with interior of the cheek

Has been corrected

Page 13: Inflammation is a consequence of subsequent phenomena caused by the imbalance of the oral cavity bacterial flora. In my oppinion this is not only the cause for inflammation as it can be induced by autoimmune disorders and many others. So it should be Inflammation may be/appear as a consequence..

Has been corrected

Page 13: exfoliative gingivitis is not the best choise in the context. It should be changed in desquamative gingivitis.

Has been corrected

Page 18: Smoking cigarettes upsets the balance. Replace upsets with causes imbalance

The content data of Figures 4 and 5 are also included in Figure 7. In my oppinion Figure 4 and 5 can be removed.

The sentences has been corrected . The figures has been removed. Figure 7 now is a figure 5.

Page 24: The conclusion chapter does not reflect data presented in the manuscript. It should be mentioned from the Introduction: Periodontal diseases are still one of the major health problems in Poland. Statistics show that out of 10 people, 9 have periodontal problems. Citation is needed.

The article needs minor revision before publication.

The sentences has been corrected and a citation has been inserted.

Conclusions chapter has been redesigned in line with the reviewer's suggestion

The chapters have been rearranged and supplemented with relevant literature citations so that they constitute a compendium for researchers and professional groups related to dentistry, and that they constitute an analysis of research achievements to date and forecasting the future direction of research.

Reviewer 3 Report

Title: The influence of traditional diet on oxidative stress and inflammation induced by bacterial biofilm in oral cavity in mammalians.

The Authors presented a review on oral anatomy, physiology, microbiology and on periodontal diseases and the microorganisms that induce them. They describe the effects of individual diet components and the role of oxidative stress in periodontal disease.

The aim of this review seems to be the analysis oral microbiota, oxidative stress and the characteristics of some foods and nutriceuticals in order to support the treatment of periodontitis, tooth decay and diseases of the oral cavity.

-The subject is relevant and interesting. The study  consist in a wide review that certainly required  to Authors great care and persistence.;

-The abstract is weak. It must be redone. It is is hard to understand, it should be put in order with results chapter reporting the results and the conclusion chapter the conclusions. It is necessary to clearly indicate (here and after in the study) which was the methodology of the study and which databases the Authors used. The aim has to be clearly reported. The results and conclusions must be clearly reported. So, it will be directly clear to whom this work will be useful.

- The whole review is not well focused on the purpose of the study so it could cause confusion. 

-This study should not be a textbook for dentists or hygienists. The references to anatomy and physiology must be reviewed, reduced and made suitable for the purpose of the study.

-It is verbose and often repetitive.  On the contrary, some major basic concepts are not correctly explained:

-It is to clear up that the pathogenic role in periodontology does not belong to a single bacteria or a limited number of periodontopathogenic bacteria. It is a highly organized ecosystem and ecosystem organiziation and pathogenicity rides on the biological and behavioral host's features. So, it is not a "true patogen" nor a "saprophyte".

-As in the study is afterward reported in the study in a disjointed way, an ecosystem has major features: they have very effectual mechanical barriers, are able to carry and dispense nutrients and gas. Moreover, they favors the exchange of recombinant plasmids. Once the ecosystem is structured it is almost impossible to change or destroy it except through professional oral hygiene (Herrera et al. 2002, Haffajee et al. 2003, Herrera et al. 2008, etc).  It is before the phase of ecosystem organization, passing from unorganized microbiota to the ecosystem, that what is reported in this study makes sense. It must be stated. 

-Oral microbiota requires an inflammatory and immune mediation to induce the periodontal disease. It is not the same in carious pathogenesis which is therefore more rapid. The clinical and therapeutic consequences are very important. This was just mentioned in the work but it needs to be pointed out. - The pathognomonic lesion of periodontal disease is the periodontal pocket. The systemic analysis of oxidative stress, etc is interesting but it is to look for literature studies deals with the inflammatory modulation in the periodontal pocket itself (e.g. -omic sciences).  A modern pathogenetic model incorporating gene, protein, and metabolite data into dynamic biologic processes is based on a multilevel framework that include disease-initiating and -resolving mechanisms that are regulated by innate and environmental factors. Thereby this study will certainly be more useful and sharp.

- The diagnosis of periodontal disease and its severity is exclusively clinical (non microbiological, biomolecular, etc.) and is carried out through periodontal charting (Tonetti & Claffey 2005, Page & Eke 2007).  Moreover, the last classification of periodontal disease is not the same reported al page 14. The correct last classification was extended during the 2017 world workshop (Papapanou et al. 2018).

The references  reported seem to be random, not in relation to what they refer: e.g. the classification reported on page 14 does not refer to references 93-109.....this classification is relating to the 1999 international workshop for a classification of periodontal diseases (Armitage 1999); at page 15  references 120-123, etc.

The cited studies (e.g  "....a study conducted at the University of Valencia...") must be reported in the references. 

The entire study has to be revised by Authors.

Author Response

Thank you very much for the apt suggestions and comments of the reviewer, which contributed to the improvement of the substantive value of our manuscript. We kindly ask you to comment on our amendments positively.

Reviewer 3

Title: The influence of traditional diet on oxidative stress and inflammation induced by bacterial biofilm in oral cavity in mammalians.

The Authors presented a review on oral anatomy, physiology, microbiology and on periodontal diseases and the microorganisms that induce them. They describe the effects of individual diet components and the role of oxidative stress in periodontal disease.

The aim of this review seems to be the analysis oral microbiota, oxidative stress and the characteristics of some foods and nutriceuticals in order to support the treatment of periodontitis, tooth decay and diseases of the oral cavity.

 -The subject is relevant and interesting. The study  consist in a wide review that certainly required  to Authors great care and persistence.;

-The abstract is weak. It must be redone. It is is hard to understand, it should be put in order with results chapter reporting the results and the conclusion chapter the conclusions. It is necessary to clearly indicate (here and after in the study) which was the methodology of the study and which databases the Authors used. The aim has to be clearly reported. The results and conclusions must be clearly reported. So, it will be directly clear to whom this work will be useful.

The abstract has been rebuilt as well as the summary chapter. The entire manuscript has been thoroughly reworked, descriptions and division of periodontal diseases have been added, and the key components of the diet, which are also the protocol for its use, have been included (page 15 and pages 13 and 14 in the manuscript), all corrections are marked in green.

We tried to present the results and conclusions as well as the purpose of the work in a clear manner.

- The whole review is not well focused on the purpose of the study so it could cause confusion. 

The entire manuscript has been thoroughly revised and rebuilt with an added paragraph on the influence of Covid-19 on the development of periodontal disease, which we believe adds to the substantive value of our work. The amendments were written to avoid additional repetition and confusion (chaos) while reading

-This study should not be a textbook for dentists or hygienists. The references to anatomy and physiology must be reviewed, reduced and made suitable for the purpose of the study.

In line with the reviewer's suggestions in the textbook for dentists and hygienists, we tried to prepare a scientific publication. References to anatomy and physiology have been reviewed, limited and adapted to the purpose of the study.

Chapters 1.1 to 1.4.8 have been removed as suggested by the reviewer and in their place new ones were written after the substantive remodeling of the entire manuscript.

Chapters 1.4.9 to 4.0 have been rearranged and included as 1 chapter together with subchapters as suggested by the reviewer.

-It is verbose and often repetitive.  On the contrary, some major basic concepts are not correctly explained

Basic concepts have been revised

-It is to clear up that the pathogenic role in periodontology does not belong to a single bacteria or a limited number of periodontopathogenic bacteria. It is a highly organized ecosystem and ecosystem organiziation and pathogenicity rides on the biological and behavioral host's features. So, it is not a "true patogen" nor a "saprophyte".

The reviewer's sentence ..” It is to clear up that the pathogenic role in periodontology does not belong to a single bacteria or a limited number of periodontopathogenic bacteria. It is a highly organized ecosystem and ecosystem organiziation and pathogenicity rides on the biological and behavioral host's features”…was included in the manuscript in chapter 1.2

-As in the study is afterward reported in the study in a disjointed way, an ecosystem has major features: they have very effectual mechanical barriers, are able to carry and dispense nutrients and gas. Moreover, they favors the exchange of recombinant plasmids. Once the ecosystem is structured it is almost impossible to change or destroy it except through professional oral hygiene (Herrera et al. 2002, Haffajee et al. 2003, Herrera et al. 2008, etc).  It is before the phase of ecosystem organization, passing from unorganized microbiota to the ecosystem, that what is reported in this study makes sense. It must be stated. 

The main features of the ecosystem are: it has very effective mechanical barriers, it is able to carry and dose nutrients and gas. Moreover, it promotes the exchange of recombinant plasmids. Once the ecosystem has been structured, it is almost impossible to change or destroy it, except through professional oral hygiene (Herrera et al. 2002, Haffajee et al. 2003, Herrera et al. 2008, etc.).

Herrera, D.; Sanz, M.; Jepsen, S.; Needleman, I.; Roldán, S. A systematic review on the effect of systemic antimicrobials as an adjunct to scaling and root planing in periodontitis patients. J Clin Periodontol. 2002;29 Suppl 3:136-59; discussion 160-2. doi: 10.1034/j.1600-051x.29.s3.8.x.

Haffajee, A. D ; Socransky, S.S.; Gunsolley, J.C Systemic anti-infective periodontal therapy. A systematic review . Ann Periodontol 2003 Dec;8(1):115-81. doi: 10.1902/annals.2003.8.1.115.

Herrera, D.;  Alonso, B.;  León, R.; Roldán, S.; Sanz, M. Antimicrobial therapy in periodontitis: the use of systemic antimicrobials against the subgingival biofilm J Clin Periodontol . 2008 Sep;35(8 Suppl):45-66. doi: 10.1111/j.1600-051X.2008.01260.x.

-Oral microbiota requires an inflammatory and immune mediation to induce the periodontal disease. It is not the same in carious pathogenesis which is therefore more rapid. The clinical and therapeutic consequences are very important. This was just mentioned in the work but it needs to be pointed out. - The pathognomonic lesion of periodontal disease is the periodontal pocket. The systemic analysis of oxidative stress, etc is interesting but it is to look for literature studies deals with the inflammatory modulation in the periodontal pocket itself (e.g. -omic sciences).  A modern pathogenetic model incorporating gene, protein, and metabolite data into dynamic biologic processes is based on a multilevel framework that include disease-initiating and -resolving mechanisms that are regulated by innate and environmental factors. Thereby this study will certainly be more useful and sharp.

The pathognomonic lesion of the periodontium is the periodontal pocket and the modulation of inflammation in the periodontal pocket itself. The modern pathogenetic model of genes, proteins and metabolites in dynamic biological processes is based on a multi-level structure that includes disease initiation and eradication mechanisms, regulated by innate and environmental factors.

This is the last line before Chapter 2.Polymorphisms in inflammatory genes such as interleukins 17A and 17F, 1, 1B and  rs1143634 and MMPs genes are associated with the risk of development of periodontitis [193-196].

  1. Slot, D.E.; Dörfer, C.E.; Van der Weijden, G.A. The efficacy of interdental brushes on plaque and parameters of periodontal inflammation: a systematic review. Int J Dent Hyg. 2008 Nov;6(4):253-64. doi: 10.1111/j.1601-5037.2008.00330.x.

193.da Silva, F.R.P.; Pessoa, L.D.S.; Vasconcelos, A.C.C.G.; de Aquino Lima, W.; Alves, E.H.P.; Vasconcelos, D.F.P. Polymorphisms in interleukins 17A and 17F genes and periodontitis: results from a meta-analysis. Mol Biol Rep. 2017 Dec;44(6):443-453. doi: 10.1007/s11033-017-4128-x. 

  1. Li, W.; Zhu, Y.; Singh, P.; Ajmera, D.H.; Song, J.; Ji, P. Association of Common Variants in MMPs with Periodontitis Risk. Dis Markers. 2016;2016:1545974. doi: 10.1155/2016/1545974.
  2. da Silva, F.R.P.; Vasconcelos, A.C.C.G.; de Carvalho França, L.F.; Di Lenardo, D.; Nascimento, H.M.S.; Vasconcelos, D.F.P. Association between the rs1143634 polymorphism in interleukin-1B and chronic periodontitis: Results from a meta-analysis composed by 54 case/control studies. Gene. 2018 Aug 20;668:97-106. doi: 10.1016/j.gene.2018.05.067.
  3. Karimbux NY, Saraiya VM, Elangovan S, Allareddy V, Kinnunen T, Kornman KS, Duff GW. Interleukin-1 gene polymorphisms and chronic periodontitis in adult whites: a systematic review and meta-analysis. J Periodontol. 2012 Nov;83(11):1407-19. doi: 10.1902/jop.2012.110655.

- The diagnosis of periodontal disease and its severity is exclusively clinical (non microbiological, biomolecular, etc.) and is carried out through periodontal charting (Tonetti & Claffey 2005, Page & Eke 2007).  Moreover, the last classification of periodontal disease is not the same reported al page 14. The correct last classification was extended during the 2017 world workshop (Papapanou et al. 2018).

 The above phrase was included before chapter 1.2.  Tonetti & Claffey 2005 were used in literature as position 197.

The most recent Classification of Periodontal Diseases has also been included as an entry as No. 19 in the literature with all its complete descriptions on pages 2 and 3.

The references  reported seem to be random, not in relation to what they refer: e.g. the classification reported on page 14 does not refer to references 93-109.....this classification is relating to the 1999 international workshop for a classification of periodontal diseases (Armitage 1999); at page 15  references 120-123, etc.

The entire literature review has been thoroughly reworked and substantially matched to the requirements of the publication

The cited studies (e.g  "....a study conducted at the University of Valencia...") must be reported in the references. 

We deleted this phrase by citing other studies from the University of Birmingham page 14 of the manuscript citation 126 and 127

The entire study has to be revised by Authors.

The entire manuscript was arranged by our team as suggested by the reviewer

Reviewer 4 Report

This paper is a very well written paper, comprehensively introducing the influence of diet on periodontal disease. However, this paper is more like a chapter of a textbook, rather than a review/research paper. There is no systematic analysis of current research achievements and prediction of the further research direction. Also this paper is not under the scope of the Materials, but more related to the journals like Dentistry Journal or Pathogens.

Author Response

Thank you very much for the apt suggestions and comments of the reviewer, which contributed to the improvement of the substantive value of our manuscript. We kindly ask you to comment on our amendments positively.

Reviewer 4

This paper is a very well written paper, comprehensively introducing the influence of diet on periodontal disease. However, this paper is more like a chapter of a textbook, rather than a review/research paper. There is no systematic analysis of current research achievements and prediction of the further research direction. Also this paper is not under the scope of the Materials, but more related to the journals like Dentistry Journal or Pathogens.

We agree that the general subject may not be covered by the scope of materials, but in our work we also describe the composition of the diet and its impact on diseases of the oral cavity, and these are the ingredients of this publication.

The chapters have been rearranged and supplemented with relevant literature citations so that they constitute a compendium for researchers and professional groups related to dentistry, and that they constitute an analysis of research achievements to date and forecasting the future direction of research.

Reviewer 5 Report

Congratulations for your work:

1) "The article is a concise compendium on periodontal diseases and the microorganisms that induce them. " this statement in the beginning of the abstract is not true. You review all diseases of microbial aetiology in the oral cavity.

2) "1.4.5. Subgingival surfaces of teeth, epithelium of fissures and subgingival pockets A healthy gingival fissure" In the previous section there is a number of wring terms, there is no such a thing as epithelium of fissures, or sub gingival pockets, please adapt based on glossary of periodontal terms throughout the text...

3) You may want to reduce the size of your manuscript be leaving out all the anatomy and physiology except for the one that is pertinent to oxidative stress and inflam-mation...1.1 to 1.4.8 to me seems irrelevant unless you want to write a book chapter.

4) From 1.4.9 up to 4 you need to summarise into 1 section and then start talking about diet. This requires a lot of work.

Author Response

Thank you very much for the apt suggestions and comments of the reviewer, which contributed to the improvement of the substantive value of our manuscript. We kindly ask you to comment on our amendments positively.

Reviewer 5

Congratulations for your work:

1)"The article is a concise compendium on periodontal diseases and the microorganisms that induce them. " this statement in the beginning of the abstract is not true. You review all diseases of microbial aetiology in the oral cavity.

The statement has been corrected. and its description is provided below

The article is a concise compendium on the etiology of pathogenic microorganisms of all oral disease inducing complexes.

2) "1.4.5. Subgingival surfaces of teeth, epithelium of fissures and subgingival pockets A healthy gingival fissure" In the previous section there is a number of wring terms, there is no such a thing as epithelium of fissures, or sub gingival pockets, please adapt based on glossary of periodontal terms throughout the text...

The chapter 1.4.5. have been removed as suggested by the reviewer. The terminology was tested and included throughout the manuscript.

3)You may want to reduce the size of your manuscript be leaving out all the anatomy and physiology except for the one that is pertinent to oxidative stress and inflam-mation...1.1 to 1.4.8 to me seems irrelevant unless you want to write a book chapter.

Chapters 1.1 to 1.4.8 have been removed as suggested by the reviewer and in their place new ones were written after the substantive remodeling of the entire manuscript.

4) From 1.4.9 up to 4 you need to summarise into 1 section and then start talking about diet. This requires a lot of work.

Chapters 1.4.9 to 4.0 have been rearranged and included as 1 chapter together with subchapters as suggested by the reviewer.

Round 2

Reviewer 3 Report

In this study an effectual link between systemic analysis of oxidative stress, etc. and the pathogenesis of the periodontal pocket, which is the pathognomonic lesion of periodontal disease, is lacking. The control of the pocket inflammatory network, the analysis of the scavenging system of oxidative molecules from the pocket and healthy gingiva or, at most, inhering the gingivocrevicular fluid (GCF) can be enlightening on the pathogenetic and therapy of periodontal disease (e.g. -resolving mechanisms). On the contrary, the study has no real clinical significance. It would have the clinical value of a food advertising since from a clinical point of view the therapies in periodontology are aimed at preventing or in any case controlling the development of periodontal pockets. Only to help to draw a scheme of analysis several studies werea aimed to analyze the inflammatory proteomic and/or cytochynes network of the periodontal pocket or GCF (e.g. Beiler et al. 2020, Martinez-Villa et al. 2020, Bostanci & Belibasakis 2018, Monari et al. 2015, Bertoldi et al. 2013, Vankeerberghen et al. 2005), the relationships between systemic inflammation and GCF (e.g. Zekeridou et al. 2019, Aral et al. 2017) and the relationship between GCF and periodontal pocket (e.g. Bertoldi et al. 2019)., Aral et al. 2017) and the relationship between GCF and periodontal pocket (e.g. Bertoldi et al. 2019). 

Author Response

Reviewer 3:

In this study an effectual link between systemic analysis of oxidative stress, etc. and the pathogenesis of the periodontal pocket, which is the pathognomonic lesion of periodontal disease, is lacking. The control of the pocket inflammatory network, the analysis of the scavenging system of oxidative molecules from the pocket and healthy gingiva or, at most, inhering the gingivocrevicular fluid (GCF) can be enlightening on the pathogenetic and therapy of periodontal disease (e.g. -resolving mechanisms). On the contrary, the study has no real clinical significance. It would have the clinical value of a food advertising since from a clinical point of view the therapies in periodontology are aimed at preventing or in any case controlling the development of periodontal pockets. Only to help to draw a scheme of analysis several studies werea aimed to analyze the inflammatory proteomic and/or cytochynes network of the periodontal pocket or GCF (e.g. Beiler et al. 2020, Martinez-Villa et al. 2020, Bostanci & Belibasakis 2018, Monari et al. 2015, Bertoldi et al. 2013, Vankeerberghen et al. 2005), the relationships between systemic inflammation and GCF (e.g. Zekeridou et al. 2019, Aral et al. 2017) and the relationship between GCF and periodontal pocket (e.g. Bertoldi et al. 2019).,

Thank you very much for all your suggestions that are very valuable to us and contribute to improving the quality of our manuscript.

Unfortunately, we did not find any of the citations cited by the reviewer in the Pub-med database, therefore we decided to present the problem more broadly in accordance with the thematic guidelines proposed by the reviewer.

- Analyze the inflammatory proteomic and/or cytokines network of the periodontal pocket or GCF

- The relationships between systemic inflammation and GCF

- The relationship between GCF and periodontal pocket

A full description of this subject is included in the sub-paragraph marked in azure.

I absolutely agree with the reviewer's suggestions, therefore new literature items have been included and described.

A literature reference search from the last few years has been included by topic name. In addition, the literature contained in the manuscript from 198 to 219 items.

Reviewer 4 Report

This paper reviewed the influence of diet on inflammation. The whole paper is significantly improved and 

Author Response

Reviewer 4:

This paper reviewed the influence of diet on inflammation. The whole paper is significantly improved

Thank you very much for all your suggestions that are very valuable to us and contribute to improving the quality of our manuscript.

Reviewer 5 Report

I would try to remove words such as "traditional" in the title ... unless there is reason for it to be there. Please do the same throughout the text. This is a language comment. I would give the text to a native speaker for edits.

Author Response

Reviewer 5:

I would try to remove words such as "traditional" in the title ... unless there is reason for it to be there. Please do the same throughout the text. This is a language comment. I would give the text to a native speaker for edits.

Thank you very much for all your suggestions that are very valuable to us and contribute to improving the quality of our manuscript.

We removed the word "traditional" from the title of the manuscript ... and applied it to the entire text. Of course, after the formal approval of the manuscript and before its final publication, we will submit the text to the native speaker for editing.
